# p53 restoration in small cell lung cancer identifies a latent cyclophilin-dependent necrosis mechanism

Jonuelle Acosta[1,2], Qinglan Li[1], Nelson F. Freeburg[1,2], Nivitha Murali[1], Alexandra Indeglia [3], Grant P. Grothusen [1,2], Michelle Cicchini[1], Hung Mai[1], Amy C. Gladstein [1,2], Keren M. Adler [1,2], Katherine R. Doerig[1,2], Jinyang Li[1], Miguel Ruiz-Torres[1], Kimberly L. Manning[1], Ben Z. Stanger [2,4], Luca Busino [1,2,4], Maureen Murphy [2,5], Liling Wan [1,2,4] & David M. Feldser [1,2,4] ✉

The p53 tumor suppressor regulates multiple context-dependent tumor suppressive programs. Although p53 is mutated in ~90% of small cell lung cancer (SCLC) tumors, how p53 mediates tumor suppression in this context is unknown. Here, using a mouse model of SCLC in which endogenous p53 expression can be conditionally and temporally regulated, we show that SCLC tumors maintain a requirement for p53 inactivation. However, we identify tumor subtype heterogeneity between SCLC tumors such that p53 reactivation induces senescence in a subset of tumors, while in others, p53 induces necrosis. We pinpoint cyclophilins as critical determinants of a p53-induced transcriptional program that is specific to SCLC tumors and cell lines poised to undergo p53-mediated necrosis. Importantly, inhibition of cyclophilin isomerase activity, or genetic ablation of specific cyclophilin genes, suppresses p53-mediated necrosis by limiting p53 transcriptional output without impacting p53 chromatin binding. Our study demonstrates that intertumoral heterogeneity in SCLC influences the biological response to p53 restoration, describes a cyclophilin-dependent mechanism of p53-regulated cell death, and uncovers putative mechanisms for the treatment of this most-recalcitrant tumor type.

Small cell lung cancer (SCLC) is the most lethal subtype of lung cancer and comprises ~15% of all lung cancer cases[1,2]. The long-standing standard of care treatment for SCLC has been platinum-based chemotherapy, which leads to significant tumor shrinkage in up to 80% of patients[3]. However, most patients later present with a recurrence of disease and succumb to treatment-resistant tumors and metastases. Recent studies have unveiled that SCLC is not a homogeneous disease and is instead comprised of four distinct molecular subtypes based on expression of lineage-defining transcription factors: *ASCL1* (SCLC-A), *NEUROD1* (SCLC-N), *POU2F3* (SCLC-P), or *YAP1* (SCLC-Y)[4]. These molecular subtypes have also been associated with distinct subtype-specific therapeutic vulnerabilities that can be used in combination with traditional chemotherapeutic treatment to prevent disease recurrence and enhance survival[5–9]. Cell types of origin influence tumor evolution and metastatic progression, further contributing to intertumoral SCLC heterogeneity[10]. These insights highlight the need

[1]Department of Cancer Biology, Perelman School of Medicine, University of Pennsylvania, Philadelphia, PA, USA. [2]Cell and Molecular Biology Graduate Group, Perelman School of Medicine, University of Pennsylvania, Philadelphia, PA, USA. [3]Biochemistry and Molecular Biophysics Graduate Group, Perelman School of Medicine, University of Pennsylvania, Philadelphia, PA, USA. [4]Abramson Family Cancer Research Institute, University of Pennsylvania, Philadelphia, PA, USA. [5]Program in Molecular and Cellular Oncogenesis, The Wistar Institute, Philadelphia, PA, USA. ✉e-mail: dfeldser@upenn.edu

to better understand how SCLC heterogeneity influences etiology of the disease, which includes the near-universal persistent selective requirement for key cancer driving mutations[11].

Together with the retinoblastoma (Rb) tumor suppressor, p53 is inactivated in the vast majority (>90%) of SCLC cases[11]. p53 is a pleiotropic tumor suppressor gene that responds to a wide array of genotoxic and cellular stresses to effectuate diverse tumor suppressive programs such as activation of apoptosis and senescence, as well as metabolic programs that may be essential for tumor suppression[12,13]. Identifying the contexts in which p53 induces one tumor suppressive program over another remains a challenge, but may hold the key to establishing novel, contextual, therapeutic strategies. Mouse models of cancer that allow for the conditional restoration of p53 function after tumor formation have identified tissue-dependent contexts where p53 preferentially activates cytostatic or cytotoxic programs[14]. In spontaneous models of B or T-cell lymphoma, p53 restoration induces canonical caspase-dependent apoptosis and rapid tumor regression[15,16]. In contrast, restoration of p53 in soft tissue sarcomas and hepatocellular carcinoma induces senescence which is followed by immune-mediated tumor clearance[16,17]. Similarly, p53 restoration in a model of Kras-driven lung adenocarcinoma also induces a senescence response followed by immune-mediated clearance, but only in advanced tumor lesions[18–20]. These studies demonstrate that p53 controls distinct tumor suppressive programs in specific cancer types, highlight the need to better understand how p53 mediates tumor suppression in distinct contexts, and illustrate the power of gene-reactivation strategies to elucidate latent mechanisms of tumor suppression.

Here we uncover multiple cancer cell autonomous roles of p53 that are selectively-activated in distinct SCLCs. We demonstrate that p53 can drive a canonical senescence program in a subset of SCLC while inducing a non-canonical form of necrotic cell death in others. This cell death program is associated with a distinct transcriptional output after p53 reactivation that was unexpectedly dependent on select members of the cyclophilin family of peptidyl proline isomerases. Importantly, without affecting DNA binding directly, cyclophilin inhibition selectively limited p53-target gene expression specifically in SCLCs that are fated to undergo necrotic cell death after p53 restoration. Additionally, we present evidence that distinct SCLC tumor subtypes may underlie the divergent cell fates that occur after p53 restoration.

## Results

### p53 restoration limits growth and metastatic spread of autochthonous SCLC

To gain insight into the requirement of sustained p53 inactivation in SCLC, we employed a *Trp53*$^{XTR}$ allele that we developed to turn off p53 expression during tumor development and then restore p53 expression in established, autochthonous cancers in the mouse[21]. *Rb1*$^{flox/flox}$;*Trp53*$^{flox/flox}$;*Rbl2*$^{flox/flox}$ (*RPR2*) mice form the basis of a model of SCLC that recapitulates the salient genetic and histopathological features of this disease[22]. In *RPR2* mice, transduction of lung epithelial cells with adenoviral vectors expressing Cre recombinase deletes *Rb1*, *Trp53*, and *Rbl2*, which results in multiclonal, aggressive SCLC after ~28 weeks[22,23]. We replaced the *p53*$^{flox}$ allele in the *RPR2* model with the *p53*$^{XTR}$ allele to generate an *Rb1*$^{flox/flox}$; *Trp53*$^{XTR/XTR}$; *Rbl2*$^{flox/flox}$ SCLC mouse model (*RP*$^{XTR}$*R2*). *Trp53*$^{XTR}$ allows for Cre-dependent inversion of an integrated gene trap to inactivate *p53* gene expression (*p53*$^{TR}$)− a state that is marked by expression of EGFP that is integrated into the gene trap. Once tumors are established, *p53* gene expression can be restored via a FlpO-dependent deletion of the gene trap (*p53*$^{R}$) (Fig. 1a). To control FlpO recombinase activity, we incorporated a *Rosa26*$^{FlpO-ER}$ allele into the *RP*$^{XTR}$*R2* model[24]. We endotracheally delivered adenoviral CMV-Cre (Ad:CMV-Cre) to a cohort of *RPR2* and *RP*$^{XTR}$*R2* mice to induce the simultaneous inactivation of endogenous *Rb1*, *Trp53*, and *Rbl2* (Fig.1b).

The resultant *RPR2* and *RP*$^{TR}$*R2* tumors were indistinguishable in size, overall burden, and histological appearance (Fig. 1c−e). All tumors had small round cells with hyperchromatic nuclei, high nuclear:cytoplasm ratios, and expressed neuroendocrine markers (*e.g.* ASCL1, CGRP, UCHL1), consistent with established features of SCLC (Fig. 1c)[25].

To determine the impact of restoring p53 expression in SCLC, we administered tamoxifen starting 25–28 weeks after tumor initiation (Fig. 1f). Consistent with the design of the *Trp53*$^{XTR}$ allele, *RP*$^{TR}$*R2* tumors expressed EGFP, whereas tumors from tamoxifen-treated animals (*RP*$^{R}$*R2*) lost EGFP expression (Fig. 1f). Reactivation of p53 expression significantly enhanced overall survival in *RP*$^{R}$*R2* animals compared to vehicle control-treated *RP*$^{TR}$ *R2* cohorts or tamoxifen-treated *RPR2* cohorts (Fig. 1g). Moreover, metastatic nodules in the liver, a common site of metastatic spread in human SCLC, were significantly fewer in *RP*$^{R}$*R2* animals (Fig. 1h, i)[26]. Notably, *RP*$^{R}$*R2* animals eventually succumbed to SCLC burden, but these tumors were GFP+, indicating that p53 restoration failed in a subset of cells which then expanded over time (Supplementary Fig. 1). Nevertheless, the extended survival after p53 restoration suggested that p53 limits SCLC growth or disease progression.

### p53 restoration induces either senescence or necrosis in SCLC in vivo

To determine how p53 restoration influences tumor maintenance, we tracked tumor size over time using micro computed tomography (µCT). We observed that the mice with restored p53 expression had significantly less total volume of lung tumors relative to control animals after two weeks of restoration (Fig. 2a). In control mice, individual tumors all increased in volume over the two-week observation period (1.3 to 378% growth). However, p53 restoration had a heterogenous effect on individual tumors, where approximately half had limited growth relative to controls (3.6 to 98% growth), and the remainder regressed relative to their starting size (7.2% to 59.5% reduction) (Fig. 2b). These data suggest that SCLC may have significant intertumor heterogeneity that impacts the potency of tumor suppression and/or the selection of specific tumor suppressive programs induced by p53 restoration.

Canonical p53-controlled tumor suppressive programs are senescence and apoptosis. Both of these fates have previously been observed in cancer models with reversible p53 inactivation[15–19]. Consistently, we found that p53 reactivation induces a significant decrease in cell proliferation across all tumors but leads to the progressive accumulation of senescence-associated-β-galactosidase (SA-β-Gal) activity in only a subset (Fig. 2c−e). However, p53 reactivation did not increase markers of apoptosis after 3 or 14 days, despite significant tumor regression (Fig. 2f, g). Instead, we found a preponderance of necrotic and fibrotic areas in a subset of SCLC tumors two weeks after p53 reactivation (Fig. 2h, i). These findings suggest the possibility of intertumor heterogeneity wherein p53 reactivation leads to the induction of reduced proliferation with features of cellular senescence in some tumors and necrotic form of cell death in others.

### p53 induces a form of cyclophilin-dependent necrosis in SCLC

To better understand the distinct tumor suppressive programs induced by p53 in SCLC, we generated cancer cell lines from individual tumors from *RP*$^{TR}$*R2* mice prior to p53 restoration (Fig. 3a). Consistent with their SCLC origin, all cell lines initially expressed the neuroendocrine marker ASCL1 and lacked expression of RB (Supplementary Fig. 2a, b). Exposure of these cells to 4-hydroxytamoxifen (4-OHT) to activate FlpO-ER activity led to the conversion of the *Trp53*$^{TR}$ allele to the *Trp53*$^{R}$ state (Supplementary Fig. 2c). Likewise, EGFP reporter gene expression was diminished in concordance with the emergence of p53 protein expression and expression of the canonical p53-target gene p21 (Fig. 3b, Supplementary Fig. 2d). Consistent with other p53-deficient cancer models, all SCLC tumor-derived cell lines had high

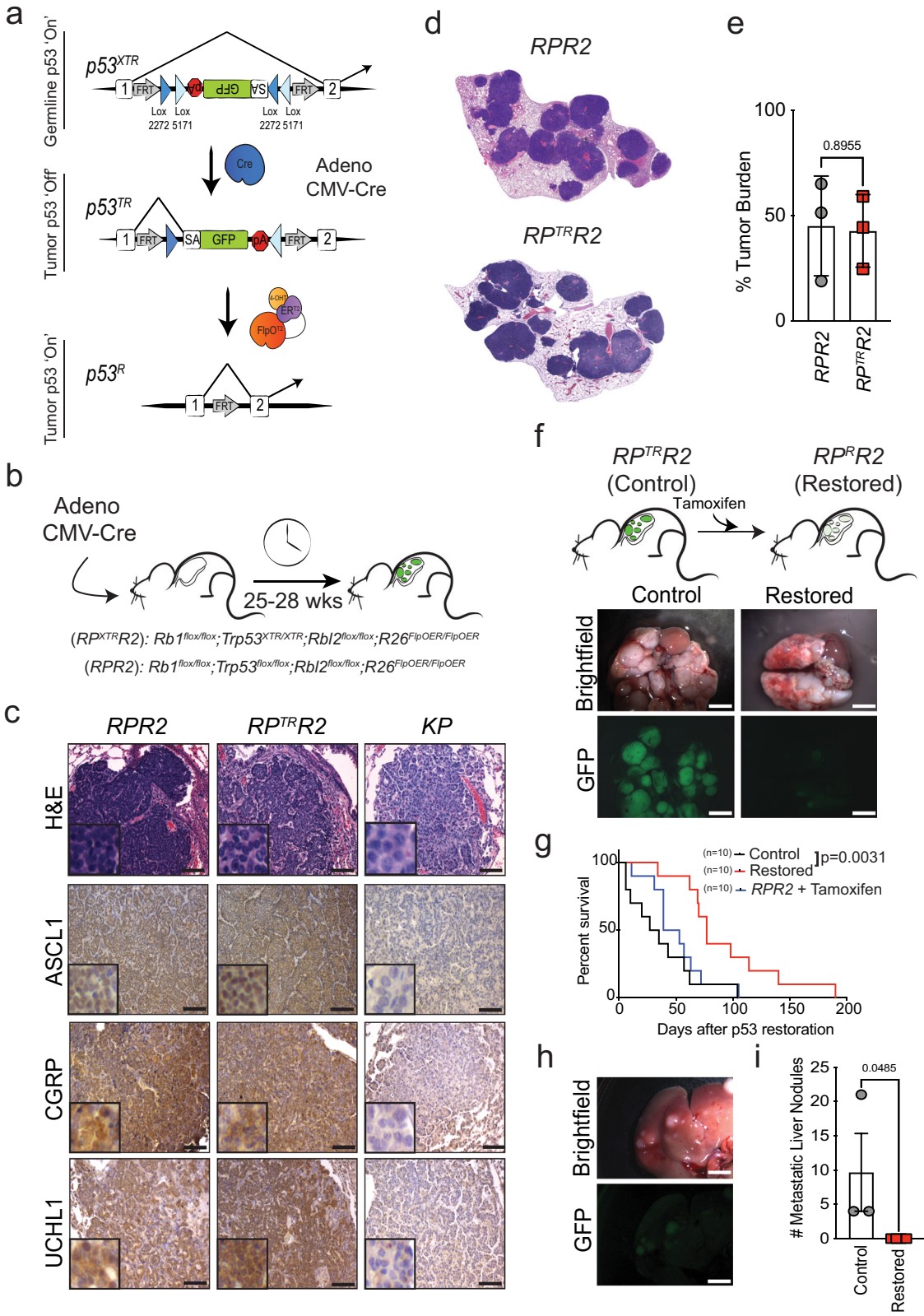

levels of the $p19^{Arf}$ tumor suppressor, a negative regulator of the p53-targeting ubiquitin ligase, *Mdm2* (Supplementary Fig. 2e)[27]. Most importantly, and consistent with our in vivo findings, p53 reactivation induced two disparate effects across distinct cell lines (Fig. 3c, Supplementary Fig. 3). A subset (4 of 8, 50%) of the cell lines remained viable after p53 reactivation (hereafter "Type V") but halted the cell cycle and induced SA-β-Gal activity (Fig. 3f–h, Supplementary Fig. 4a,

b). In contrast, the remaining subset of cell lines (4 of 8, 50%) underwent a sudden cell lysis (necrosis) approximately 3 to 5 days after p53 reactivation (hereafter "Type D") (Fig. 3c–e, Supplementary Movie 1). Cell death was not induced by tamoxifen treatment per se, or differences in p53 expression between cell lines (Supplementary Fig. 4c, d). Similar to in vivo observations, we detected no markers of apoptosis in any of the Type D or Type V cell lines at *'early'* (24 h) or *'late'* (72 h)

**Fig. 1 | p53 restoration limits SCLC growth and metastasis in vivo. a** Schematic of the *p53^XTR* allele. Top, *p53^XTR* (expressed): XTR gene trap cassette consists of a splice acceptor (SA), enhanced Green Fluorescent Protein (GFP), and the polyadenylation transcriptional terminator (pA). Stable inversion is achieved by the use of two pairs of mutually incompatible mutant *loxP* sites (Lox2272 and Lox5171) arranged in the 'double-floxed' configuration. Middle, *p53^TR* (trapped): inhalation of Cre-expressing adenoviral vectors induces permanent gene trap inversion and conversion to the *p53^TR* allele inactivating *Trp53* expression. Transcripts are spliced from exon 1 to the GFP reporter. Bottom, *p53^R* (restored): the *Rosa26^FlpO-ERT2* allele enables tamoxifen-dependent conversion of trapped *p53^TR* to its restored *p53^R* allelic state via excision of the gene trap. **b** *RPR2* and *RP^XTR R2* mice were infected with $1.0 \times 10^8$ p.f.u of Ad:CMV-Cre to initiate tumor formation. Between 25–28 weeks after tumor initiation, mice were treated with vehicle or tamoxifen, as described. **c** H&E and IHC for the neuroendocrine markers, ASCL1, CGRP, and UCHL1 in *RPR2*, *RP^TR R2* and *KP*

mice. Scale bars: 25 μm for H&E, 25 μm for IHC; insets are magnified 5×. **d** Representative scans of tumor-bearing lobes from *RPR2* or *RP^TR R2* mice. **e** Tumor burden 25–28 weeks after tumor initiation. *RP^TR R2* and *RPR2*, *n* = 3 mice. Statistical significance was determined by two-tailed Student's *t*-test. Error bars represent mean ± s.d. **f** Brightfield and fluorescent micrographs of lungs from *RP^TR R2* mice treated with vehicle (Control) or tamoxifen (Restored). Scale bars: 4.4 mm (**g**) Kaplan-Meier survival analysis. *RP^TR R2*, *n* = 10 mice per treatment group (Control or Restored); *RPR2*, *n* = 10 mice. Statistical significance determined by Mantel-Cox log rank test. **h** Brightfield and fluorescent micrographs of metastatic liver nodules from control *RP^TR R2* mice. Scale bars: 4.4 mm (**i**) Quantification of metastatic liver nodules in control (*n* = 3 mice) and restored (*n* = 3 mice) after 9–16 days of tamoxifen treatment. Error bars represent mean ± SEM. Statistical significance determined by one-tailed Student's *t*-test. Source data are provided as a Source Data file.

timepoints (Fig. 3i, j, Supplementary Fig. 4e). Furthermore, over expression of the anti-apoptotic protein, Bcl2, did not block p53-mediated necrosis in Type D cells, supporting the notion that p53 is inducing a non-apoptotic form of cell death (Supplementary Fig. 4f–i).

To gain insight into the mechanism of cell death induced by p53 reactivation in SCLC, we treated Type D cells with pharmacological inhibitors of apoptosis (ZVAD-FMK), ferroptosis (Ferrostatin-1), necropstosis (Necrostatin-1s), and cyclophilin-dependent death (Cyclosporin A). Surprisingly, only Cyclosporin A (CsA) treatment protected all Type D cell lines from p53-mediated cell death (Fig. 3k, Supplementary Fig. 4j, k). CsA is most notable for its ability to suppress T cell function by forcing a neo-protein-protein interaction with calcineurin and cyclophilin, which blocks downstream NFAT signaling[28–30]. FK506 is another small molecule that forces a distinct neo-protein-protein interaction of FKBP12 with calcineurin which also blocks NFAT signaling[29,31,32]. To distinguish whether this p53-mediated death requires NFAT activation or an alternative activity of cyclophilins themselves, we treated cells with a cyclophilin-specific inhibitor that does not affect calcineurin/NFAT signaling (NIM-811) or FK506 which blocks calcineurin/NFAT but not cyclophilin activity. NIM-811, but not FK506, blocked cell death similarly to CsA, confirming that cyclophilins are key effectors of p53-induced death in Type D cells (Fig. 3l, Supplementary Fig. 4l). Finally, to establish the role of cyclophilins in p53-mediated tumor regression in vivo, we tracked individual autochthonously arising tumors in live mice over time using μCT during daily CsA (15 mg/kg) treatments. CsA treatment unveiled a subset of cyclosporine-sensitive tumors that were significantly compromised in their ability to regress in response to p53 reactivation (Fig. 3m, n). Taken together, these data suggest that p53 induces a form of cell death dependent on cyclophilins in SCLC, and that inhibition of cyclophilins can limit p53-mediated tumor cell death in vitro and in vivo.

We next investigated the relationship of p53-mediated senescence and p53-mediated necrosis programs in SCLC. While p53-mediated necrosis is blocked in Type D cells treated with CsA, the cells do not express significant levels of SA-βgal activity, despite being fully arrested (Supplementary Fig. 5). Interestingly, Nutlin-3a treatment, which enhances p53 stability upon its restoration, promoted cell death induction in both Type V and Type D cells, which could be blunted by CsA treatment in each SCLC type (Supplementary Fig. 6). Taken together, these findings highlight that both SCLC subtypes are capable of inducing cyclophilin-mediated death after p53 restoration but that Type V SCLC requires deregulated control of p53 expression to do so.

### Immune infiltration is not necessary for induction of p53-mediated necrosis

Senescence following p53 reactivation is sometimes associated with immune mediated tumor cell clearance[17,20]. To establish the extent to which immune infiltration promotes tumor regression, we profiled immune cells by flow cytometry in SCLC tumors after p53 restoration.

Although widespread immune infiltration into SCLC tumors after p53 reactivation was not prominent, there was a significant increase in macrophages (CD45+, F4/80+) after p53 restoration (Supplementary Figs. 7, 8a). However, these macrophages were not localized within senescent tumors but instead were predominantly found within tumors that had large necrotic and/or fibrotic regions after p53 reactivation. This suggests that these cells are likely infiltrating SCLC tumors to repair damaged tissue and clear dead cells, instead of directly culling senescent tumor cells (Supplementary Fig. 8b). Finally, we transplanted Type D SCLC cell lines into immunocompromised (NCr-Foxn1Nu) nude mice. p53 reactivation in tumor-bearing mice induced tumor regression and necrosis in each of four independent Type D cell lines indicating that the adaptive immune system plays a minimal role in p53-mediated Type D cell death (Supplementary Fig. 8c–g). Taken together, with in vitro cell culture observations indicating that Type D cells self-destruct in the absence of any immune cells, these data suggest that the immune system is not required to trigger p53-mediated Type D cell death in vivo.

### Cyclophilins A and E are key effectors of p53-mediated necrosis in SCLC

CsA limits mitochondrial permeability transition pore (mPTP)-induced necrosis by limiting binding of p53 with cyclophilin D (encoded by *Ppif*), which disrupts mitochondria membrane permeability and induces necrosis during tissue ischemia[33]. To establish the role of specific cyclophilin(s) in p53-mediated necrosis in Type D cells, we generated LentiCRISPRv2-mCherry vectors that individually target cyclophilin family members known to interact with CsA (*Ppia, Ppib, Ppic, Ppid, Ppie, Ppif, Ppig, Ppih*)[34]. These vectors facilitate positive-enrichment assays, where knockout of critical death effectors will lead to enrichment of mCherry-positive cells after p53 reactivation (Fig. 4a, Supplementary Figs. 9, 10). We discovered that only sgRNAs targeting *Ppia* and *Ppie* were significantly enriched upon p53 reactivation, and that the enrichment was specific to Type D cells (Fig. 4b–e, Supplementary Fig. 11a, b). Moreover, the combined expression of sgRNAs targeting *Ppia and Ppie* suppressed p53-mediated cell death cooperatively and to a similar extent as sgRNAs targeting p53 itself (Fig. 4f, Supplementary Fig. 11c). No other sgRNA, including those targeting *Ppif*, were enriched (Fig. 4b, Supplementary Fig. 11d). While p53 interactions with cyclophilin D were minimally present in both Type V and Type D SCLC (Supplementary Fig. 12), we could not establish prominent p53 localization to the mitochondria or detect evidence for a unique impact on mitochondria physiology after p53 restoration in Type D SCLC cell lines (Supplementary Fig. 13). These data suggest that although CsA potently blocks p53-mediated cell death in Type D SCLC, the mechanism is distinct from mPTP-induced necrosis, and instead involves specific activities associated with cyclophilin A (CypA) and cyclophilin E (CypE).

How CypA and CypE regulate p53 function is not understood and cannot easily be explained by expression of either protein in Type D or

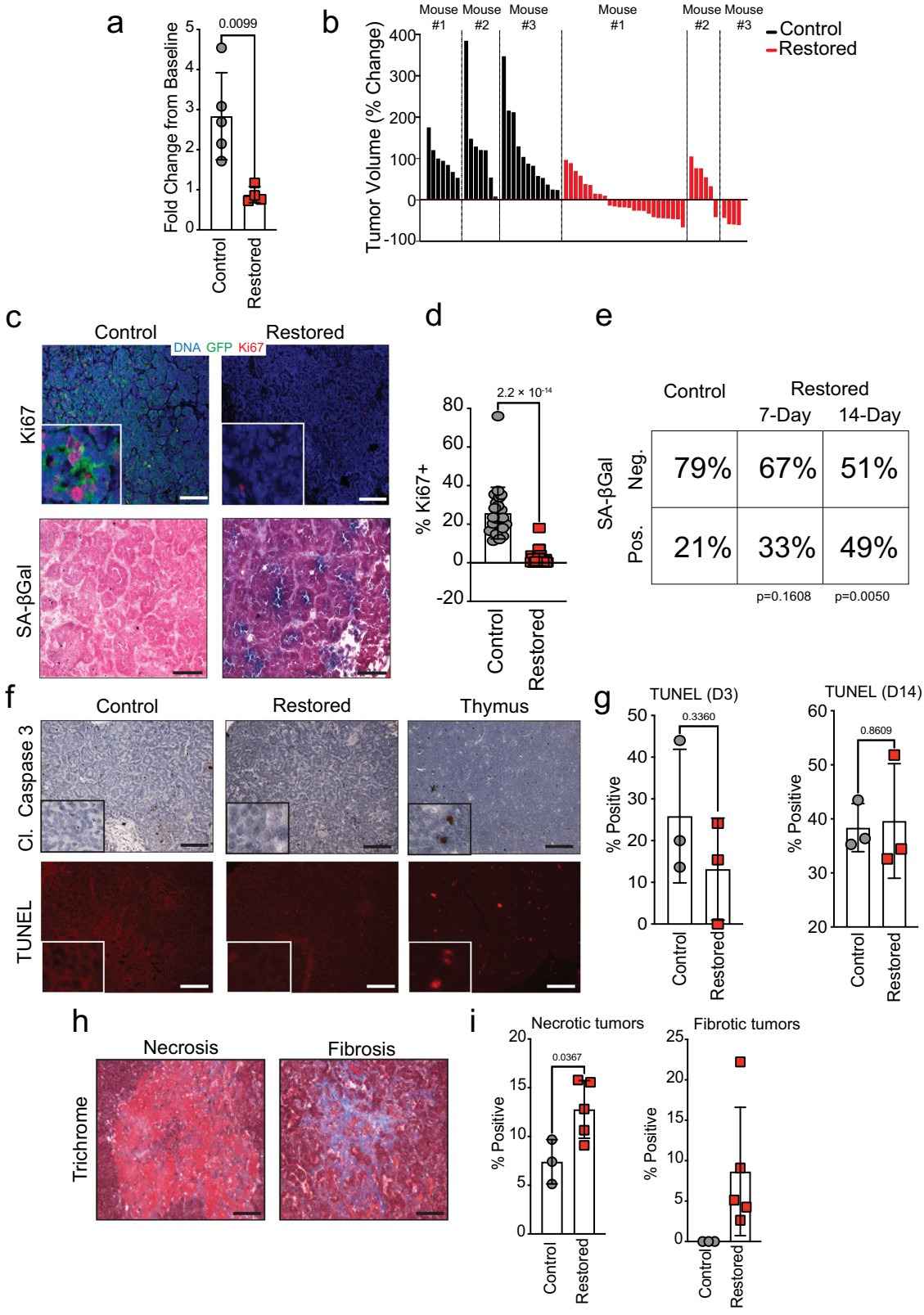

Type V cells (Supplementary Fig. 14). CypA has been shown to physically interact with p53 to affect p53 target gene selection[35]. We performed co-immunoprecipitation experiments using FLAG-PPIA or FLAG-PPIE, and HA-p53. Consistently, CypA strongly interacted with p53 (Fig. 4g). However, we observed no interaction between CypE and p53 (Fig. 4g). To elucidate whether CsA modulates p53-cyclophilin interactions, we co-immunoprecipitated FLAG-PPIA or FLAG-PPIE, and

HA-p53 in the presence or absence of CsA. Strikingly, the CypA-p53 interaction was completely inhibited by CsA treatment (Fig. 4h). We validated the endogenous occurrence of a Cyclophilin-p53 interaction using proximity ligation assays. Consistent with the co-immunoprecipitation experiments, CypA interacted with p53 whereas CypE did not. The CypA-p53 interaction occurred predominantly in Type D cells but also to a lesser extent (-10-fold less) in

**Fig. 2 | p53 restoration induces either senescence or necrosis in SCLC in vivo.**
**a** Tumor burden analysis from μCT scans. Fold change from baseline measurements. Control includes *RPR2* (*n* = 3) and *RP^{TR}R2* (*n* = 2) mice. Restored *RP^RR2* (*n* = 5) mice after 11–14 days of treatment. Statistical significance determined by two-tailed Student's *t*-test. Error bars represent mean ± s.d. **b** Waterfall plot of individual tumor volume plotted as percentage change from baseline in control (*n* = 3 *RP^{TR}R2*) and restored (*n* = 3 *RP^RR2*) mice. **c** Photomicrographs for GFP and Ki67 immunofluorescence and SA-β-Gal stained *RP^{TR}R2* tumors 7 days after vehicle (Control) or tamoxifen (Restored) treatment. Scale bars: 25 μm for SA-β-Gal, 50 μm for IF; insets are magnified 5×. **d** Quantification of Ki67 staining in control (*n* = 23) and restored (*n* = 35) tumors from (**c**). Statistical significance determined by two-tailed Student's *t*-test. Error bars represent mean ± s.d. **e** Contingency analysis of SA-β-Gal staining from (**c**); *n* = 61 tumors from 7 Control mice; *n* = 63 tumors from 4 Restored mice at

*t* = 7 days; *n* = 41 tumors from 4 Restored at *t* = 14 days. Statistical significance determined by two-tailed Fisher's exact test. **f** Photomicrographs for cleaved Caspase-3 IHC or TUNEL stained *RP^{TR}R2* tumors. Scale bars: 25 μm for IHC, 50 μm for TUNEL; insets are magnified 5×. **g** Quantification of TUNEL staining from (**f**) 3 or 14 days after tamoxifen treatment; *n* = 3 mice for all treatment groups. Statistical significance was determined by two-tailed Student's *t*-test. Error bars represent mean ± s.d. **h** Representative photomicrographs for necrosis or fibrosis positive *RP^{TR}R2* tumors from trichrome stained tissue sections. Scale bars, 25 μm. **i** Quantification of trichrome staining from (**h**) after 14 days of tamoxifen treatment. *n* = 3 mice for control mice; *n* = 5 mice for restored mice. Statistical significance determined by two-tailed Student's *t*-test. Error bars represent mean ± s.d. Source data are provided as a Source Data file.

Type V cells. Moreover, regardless of cell type, CsA treatment abrogated CypA-p53 interactions (Fig. 4i, j). Interestingly, the CypA-p53 interactions were predominantly localized to the nucleus with some interactions occurring in perinuclear regions (Supplementary Movie 2). These data suggest that CypA could be directly modulating p53 function in the nucleus and that disrupting the CypA-p53 interaction with CsA compromises the ability of p53 to induce cell death in Type D SCLC.

## Type V and Type D cells have features of distinct molecular subtypes of human SCLC

Cellular context influences p53 function to induce distinct tumor suppressive programs[36–40]. Thus, we conducted RNA-sequencing in Type D and Type V SCLC cells to gain insight into context-specific differences that may differentially regulate p53 function. Principal component analysis (PCA) revealed that Type V or Type D class is the primary distinguishing feature among all samples, making up >47% of the variation in the data (Fig. 5a). Secondly, p53 restoration status made up >12% of the variation in the data (Control versus Restored). To further characterize the baseline transcriptional differences (Control vs Control) between Type D and Type V SCLC cells, we conducted differential gene expression and gene set enrichment analysis (GSEA)[41,42]. While only 109 genes were significantly enriched in Type D cells and 57 genes significantly enriched in Type V cells (Fig. 5b), multiple gene sets distinguished Type V and Type D cells. Notably, Type D cells were enriched for metabolic gene signatures, whereas Type V cells were enriched for gene signatures regulating epithelial mesenchymal transition, Wnt and hedgehog signaling, and inflammatory processes (Fig. 5c). Human SCLC is a highly heterogeneous disease that has recently been stratified into four molecular subtypes based on expression of lineage defining factors: ASCL1 (SCLC-A), NEUROD1 (SCLC-N), YAP1 (SCLC-Y), POU2F3 (SCLC-P)[4]. As such, we determined whether Type D and Type V cells were enriched for human SCLC-subtype-specific gene signatures[43]. We generated gene signatures associated with Type D and Type V cells based on differentially regulated genes identified above (Fig. 5b) and used SCLC-CellMiner to compare each signature to human SCLC cell line gene expression datasets[44]. SCLC-A human cells are significantly enriched for the Type D signature, whereas SCLC-N human cells are enriched for the Type V signature (Fig. 5d, e). These gene expression correlations suggest that Type D and Type V SCLC are defined by basal gene expression related to two molecular subtypes of human SCLC.

To gain functional insight into the context-dependent role of p53 in human SCLC, we used adenoviral vectors to re-introduce wild-type p53 into a panel of five human SCLC cell lines (Fig. 5d–o). H2081 and H2195 cells are both ASCL1-high and highly enriched for the Type D signature. Similar to mouse Type D cell lines, expression of p53 reduced their viability (Fig. 5f, g, k, l). In H2081 cells, the loss in cell viability due to p53 re-expression was limited by treatment with CsA; CsA treatment in H2195 cells, however, had no effect. In contrast, re-expression of p53 in HCC4004, HCC2433, and H69 had no measurable

impact on their viability (Fig. 5h–j, 5m–o). Interestingly, H69 cells were ASCL1-high but were poorly enriched for the Type D signature, whereas HCC4004 and HCC2433 were significantly enriched for the Type V-signature. While additional studies need to be conducted to further understand the role of SCLC subtype in the response to p53 in human SCLC, our findings suggest that intertumoral heterogeneity in both mouse and human SCLC influences the tumor suppressive programs induced by p53 with at least a subset of human SCLC vulnerable to p53-mediated, cyclophilin-dependent cell death.

## SCLC heterogeneity does not dictate p53 binding

To examine p53 DNA binding patterns, we conducted chromatin immunoprecipitation (ChIP)-sequencing 48 h after p53 reactivation in Type D and Type V cells. As a proof of principle, we observed high p53 ChIP-signal (Restored) and very low noise (Control) at the prototypical p53 target gene, *Cdkn1a* (Supplementary Fig. 15a). Across the genome, the location of p53 binding was largely similar across all SCLC cell lines; however, Type V cells had a significantly higher degree of p53 binding at most sites (Supplementary Fig. 15b–15d). Differential peak calling analysis identified 1032 peaks in Type V cells which corresponded to 508 differentially bound genes (Type V DBGs). Only 3 peaks had significantly higher binding in Type D cells, each of which corresponded to a specific gene (Type D DBGs) (Supplementary Fig. 15e). However, differential peak calling between Type D and Type V cells did not reflect distinct changes in gene expression after p53 restoration. In fact, 434 out 508 Type V DBGs were not differentially expressed at all after p53 restoration in Type V or Type D cells. Moreover, for those 74 Type V DBGs that were differentially expressed after p53 restoration, the vast majority were also similarly up- or down-regulated in Type D cells (Supplementary Fig. 15f–15h). Expression of the 3 Type D DBGs was likewise not distinctly up- or down-regulated by p53 in Type D or Type V cells (Supplementary Fig. 15i). These data suggest that differential p53 binding is not a robust predictor of the distinct physiological programs orchestrated by p53 in Type D and Type V SCLC, and that p53 binds to promoter-proximal sequences within a similar repertoire of genes in both Type D and Type V cells.

## p53 induces a distinct transcriptional program in type D SCLC that is cyclophilin-dependent

We conducted differential gene expression analysis between control and p53-restored Type D and Type V cells to determine whether p53 regulated distinct transcriptional programs in each. We identified 764 genes that were differentially expressed (*p* < 0.05 and log₂ fold change ≥1) after p53 reactivation in Type D cells and 250 genes that were differentially expressed in Type V cells 72 h after p53 restoration (Supplementary Fig. 16a, b). Importantly, *Trp53* transcript abundance and the induction of p53 Hallmark Genes were similarly up-regulated after p53 restoration, suggesting that regulation of the *Trp53* gene per se or p53's regulation of canonical target genes were not responsible for disparate cell fates after p53 restoration in Type D and Type V cells (Supplementary Fig. 16c, d). Taking the union of the p53-regulated genes identified

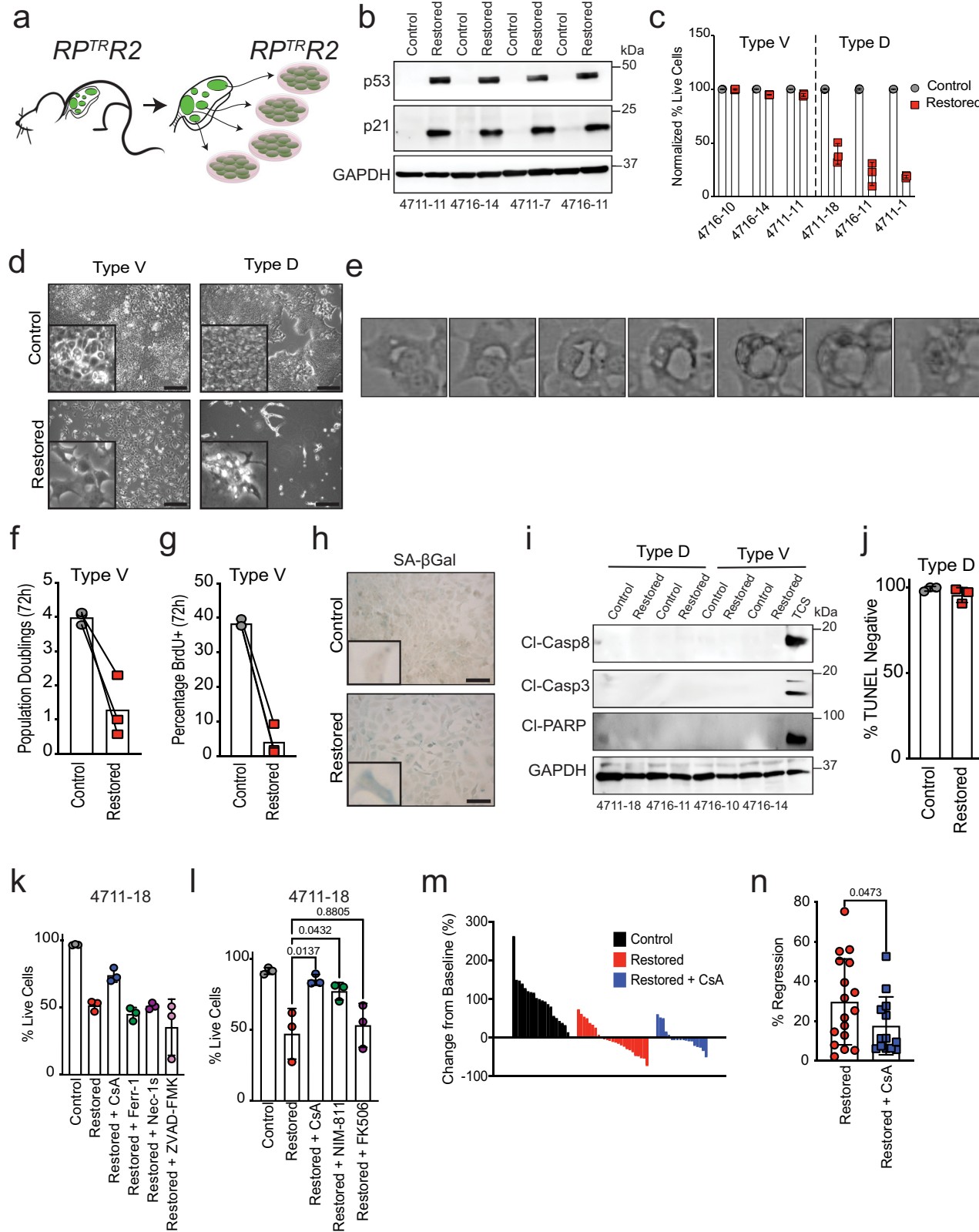

in Type D and Type V cells, we performed unbiased hierarchical-clustering analysis in control and p53-restored samples. The genes clustered into 6 distinct gene expression modules (Supplementary Data 1). Generally, the expression patterns of all genes in each Module moved in the same direction after p53 restoration in both cell types regardless of whether the gene was identified in Type V or Type D samples (Fig. 6a, Modules 1–4, and 6). However, the 102 genes in Module

5 were disproportionally elevated in Type D cells after p53 restoration compared to a negligible induction in Type V cells (Fig. 6a). Interestingly, the 102 genes fell into multiple gene ontology (GO) and functional enrichment categories associated with regulation of cell death processes (Supplementary Fig. 16e–j, Supplementary Data 1). These data suggest that p53 may be distinctly regulating a small subset of genes to induce distinct tumor suppressive programs in Type D SCLC.

**Fig. 3 | p53 reactivation induces cellular senescence or cyclophilin-dependent death. a** RP$^{TR}$R2 mice were infected with $1.0 \times 10^8$ p.f.u of Ad:CMV-Cre to initiate tumor formation. Between 25–28 weeks after tumor initiation, tumors were excised for cancer cell line establishment. **b** Immunoblot analysis of RP$^{TR}$R2 cell lines ($n = 4$) for p53 and p21 expression 24hrs after 4-OHT treatment. GAPDH is a loading control. **c** Flow cytometry-assisted cell viability assay in representative Type V (4711-11, 4716-10, 4716-14) and Type D (4711-1, 4711-18, 4716-11) cells 5 days after 4-OHT treatment. Live cell percentage determined by quantification of DAPI negative population; $n = 3$ technical replicates for all cell lines and treatment groups. Error bars represent mean ± s.d. **d** Brightfield photomicrographs from Type V and Type D cell lines after 5 days of 4-OHT treatment. Scale bars, 250 μm. **e** Time lapse video of a Type D cell dying after p53 reactivation. Video accounts for 72-hour time span between 24 and 96 h after p53 restoration. **f** Quantification of cell population doublings 72 h after 4-OHT treatment. Each symbol represents the mean of $n = 3$ technical replicates per Type V cell line (4711-11, 4716-10, 4716-14). **g** Quantification of BrdU+ cells 72 h after 4-OHT treatment. Symbols represent independent Type V cell lines ($n = 3$; 4711-11, 4716-10, 4716-14). **h** Brightfield photomicrographs of SA-β-Gal stained Type V cells representative of three independent Type V cell lines. Scale bars, 25 μm; insets are magnified 5×. **i** Immunoblot analysis for apoptosis markers in Type V (4716-10, 4716-14) and Type D (4711-18, 4716-11) cells 4 days after 4-OHT treatment. Positive control (TCS) treated with TNF-α (1 μg/mL), Smac mimetic (SM-164; 100 nM), and cyclohexamide (10 μg/mL) for 8 h. GAPDH is loading control.

**j** Quantification of flow cytometry-assisted TUNEL staining in Type D ($n = 3$) cells. Symbols represent independent Type D cell lines (4711-1, 4711-18, 4716-11). Each symbol represents the mean of $n = 3$ technical replicates per Type D cell line. Error bars represent mean ± s.d. **k** Type D (4711-18) cells were treated with 4-OHT and specific cell death inhibitors for 72 h. Percentage of live cells (DAPI-) were determined using flow cytometry. Each symbol represents a technical replicate ($n = 3$). Statistical significance was determined by two-tailed Student's $t$-test. Error bars represent mean ± s.d. Experiment was conducted in $n = 2$ independent Type D cell lines. **l** Type D (4711-18) cell line was treated with 4-OHT and cyclophilin (CsA, NIM-811) or FKBP (FK506) inhibitors for 72 h. Percentage of live cells (DAPI-) were determined using flow cytometry. Each symbol represents mean of technical replicates ($n = 3$). Statistical significance was determined by one-way ANOVA followed by Dunnett's multiple comparison test. Error bars represent mean ± s.d. Data represent $n = 3$ independent experiments. **m** Individual tumor volume analysis using μCT. Waterfall plot percentage change from baseline in control ($n = 3$), restored ($n = 3$), and restored + CsA ($n = 5$) mice after 15 days of treatment. **n** Quantification of percentage regression of tumors in (**m**). $n = 17$ regressing tumors from $n = 3$ mice treated with tamoxifen; $n = 13$ regressing tumors from $n = 3$ mice treated with tamoxifen and CsA. Error bars represent mean ± s.d. Statistical significance determined by one-tailed Student's $t$-test. Source data are provided as a Source Data file.

That cyclophilin activity is required for p53-mediated Type D SCLC cell death and that p53 interacts with CypA in a CsA dependent manner led us to hypothesize that cyclophilins may be required for p53-dependent transcription in this context. Thus, we conducted ChIP- and RNA- sequencing experiments after p53 restoration while treating SCLC cells with CsA. Surprisingly, CsA treatment did not significantly influence p53 binding across the genome in Type D or Type V cells (Fig. 6b–e). Moreover, CsA treatment did not alter gene expression prior to p53 restoration in any cell line, including expression of cyclophilin family members (Supplementary Fig. 17a–c). To explore the effect of CsA on p53-induced gene expression, we conducted GSEA and compared how CsA modulated p53 transcriptional output between Type D and Type V cells. Strikingly, CsA treatment abolished the enrichment of Hallmark p53 Pathway genes in Type D cells after p53 restoration. However, the effect of CsA on Hallmark p53 Pathway genes in Type V cells had the reverse effect, slightly promoting expression of these genes after p53 restoration (Fig. 6b–f). Consistently, after p53 restoration, the average change in expression for each gene in the Hallmark p53 Pathway was significantly diminished in Type D cells in the presence of CsA treatment. However, the addition of CsA to Type V cells had no effect (Fig. 6g). Given that Module 5 genes were predominantly induced by p53 restoration specifically in Type D cells (Fig. 6a, and Supplementary Fig. 16i), we determined the extent to which CsA treatment influenced p53-mediated transcription of these genes. Strikingly, CsA abolished the induction of Module 5 genes in Type D cells (Fig. 6h). Despite the failure of p53 to induce canonical target gene expression generally and most potently at Module 5 loci in the presence of CsA in Type D cells, p53 bound equally well, if not slightly better, to promoter proximal regions within these genes (Fig. 6i, j, Supplementary Fig. 17d, e).

Although long-term CypE KO was not possible, we could isolate Type D cells where CypA levels were significantly reduced (though not absent) after CRISPR-mediated gene targeting (Supplementary Fig. 11a, b, Supplementary Fig. 18a). Therefore, we conducted RNA-sequencing experiments in Type D cells after selections of a non-targeting or *Ppia*-targeting sgRNA. Interestingly, CRISPR-targeting of CypA alone reduced the enrichment of Hallmark p53 Pathway genes and significantly depleted the expression of Module 5 genes after p53 restoration (Fig. 6k, l, Supplementary Fig. 18). These data indicate that p53 transcriptional output, and not target gene binding, requires the additional activity of cyclophilins, specifically in Type D SCLC. Moreover, p53 can transactivate a unique subset of genes (Module 5) specifically in Type D SCLC that are likely required for p53-mediated cell death in SCLC.

## Discussion

Our study provides key insight into the persistent selective requirement for p53 inactivation and the potential therapeutic responses to p53 reactivation in SCLC. Previous studies using mouse models with reversible p53 expression in distinct cancer contexts report a uniform response to p53 reactivation, with tumors either undergoing a senescence response followed by immune-mediated tumor clearance or apoptotic cell death[15–20]. We demonstrate that p53 reactivation limits SCLC disease progression by inducing one of two distinct tumor suppressive programs in individual, clonally-derived neoplasms, with approximately half experiencing a cellular senescence response not associated with a major immune reaction, and the rest undergoing cyclophilin-dependent necrotic cell death.

Cyclophilins are a class of peptidyl-proline isomerases (PPIAses) that are part of a protein superfamily composed of cyclophilins, parvulins, and FK506-binding proteins[45]. PPIAses canonically regulate protein function by inducing conformational changes after catalyzing the isomerization of proline residues[46,47]. However, further structural and functional analysis of cyclophilin family members have identified non-canonical cyclophilin functions such as RNA-binding, spliceosome regulation, and U-box ubiquitin ligase activity[34,48–51]. Unsurprisingly, this breadth of functions have implicated cyclophilins in the regulation of a plethora of biological processes and made them attractive therapeutic targets in the context of disease[33,52–54]. Previous studies establish that p53 can directly interact with Cyclophilin D (CypD encoded by *Ppif*) and Cyclophilin A (CypA encoded by *Ppia*) to potentiate mitochondrial permeability pore (mPTP) opening and necrosis, or regulate cell cycle and apoptosis function, respectively[33,35]. Our data implicate cyclophilins as key regulators of p53-mediated transcription and subsequent cell death program specifically in Type D SCLC cells. p53-mediated cell death in this context is non-apoptotic and dependent on CypA and CypE for its execution. While it remains unclear how these cyclophilin family members impact p53 transcriptional output, CypA directly interacts with p53 in the nucleus and this interaction is abolished by CsA. Moreover, CypA-p53 interactions were observed ~10-fold more frequently in Type D cells. Interestingly, CypE did not interact with p53 directly in our analysis, suggesting a possible indirect mechanism for CypE to modulate p53-mediated death. Although we do not understand how these cell contexts influence the propensity of cyclophilin-p53 interactions, we speculate that the PPIase activity of cyclophilins may directly alter the efficiency of p53 transactivation or the recruitment of specific factors that promote expression of key regulators needed to execute the cell death

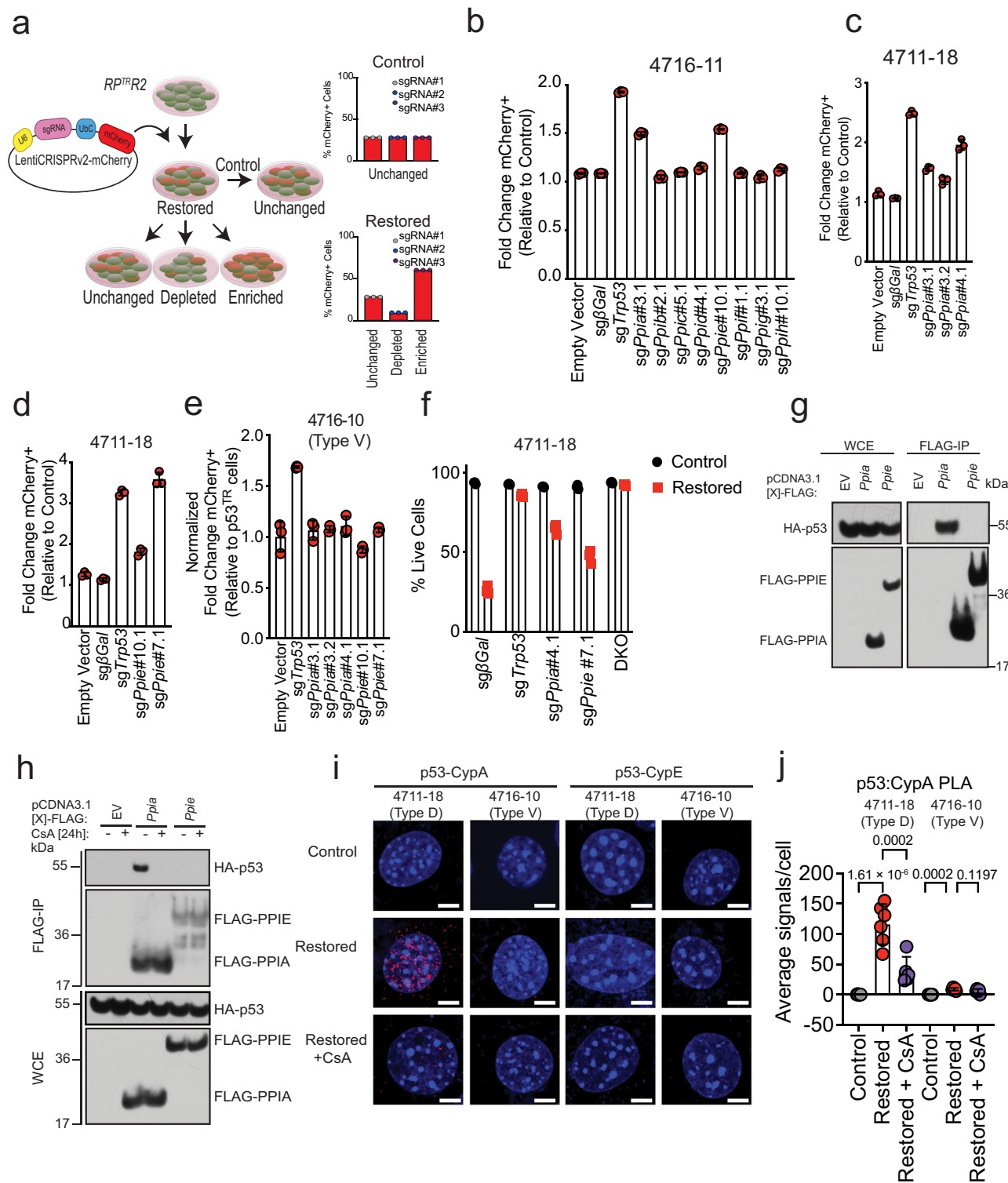

program. Future studies are required to determine how each of these cyclophilins modulate p53 function, and how they cooperate with each other or interact with cellular factors to potentiate cell death.

Comparative analysis of Type D and Type V SCLC with human SCLC gene expression datasets revealed a possible link between SCLC molecular subtype and the cellular contexts that dictate the ultimate response to p53 activity. By examining the basal transcriptional programs present in senescing (Type V) and dying (Type D) tumor-derived cells, we determined that Type D and Type V cells have gene expression programs that resemble *ASCL1* and *NEUROD1* molecular subtypes

of human SCLC, respectively. Consistent with this finding, re-introduction of wild-type p53 in ASCL1-high or NEUROD1-high human SCLC cell lines induced cell death predominantly in the ASCL1 subtype, while no measurable decrease in cell viability was observed in NEUROD1+ cells. These lineage-defining transcription factors have been shown to regulate distinct genetic programs in SCLC and disparately regulate the expression of distinct oncogenic genes to drive tumor heterogeneity[55,56]. Tumor initiation in our model was driven by adenoviral CMV-Cre, which can potentiate formation of SCLC tumors from distinct cells of origin that influence tumor evolution and

**Fig. 4 | p53-mediated Type D cell death is dependent on specific cyclophilin proteins. a** Schematic of LentiCRISPRv2-mCherry transduction of Type D cells for positive selection assay. Type V cells are infected with mCherry-expressing lentiviral vectors expressing Cas9 and sgRNAs targeting specific cell death mediators or cyclophilins. Baseline mCherry expression is measured in viable (DAPI-) control cells using flow cytometry. Upon p53 restoration, changes in the proportion of mCherry-positive cells quantified in live (DAPI-) cells. Data plotted as fold change mCherry+ population relative to Control (vehicle treated) cells. **b** Fold change in the proportion of mCherry-positive cells 4 days after 4-OHT treatment in representative Type D (4716-11) cell line expressing sgRNAs targeting distinct cyclophilins. Empty Vector, β-Gal, and a p53 targeting sgRNA used as controls. Each symbol represents a technical replicate (n = 3). Error bars represent mean ± s.d. Experiment was conducted in n = 2 independent Type D cell lines. **c**, **d** Validation of results in (**b**) using an independent Type D (4711-18) cell line and additional sgRNAs targeting exons 3 and 4 of the *Ppia* gene (**c**) or exons 7 and 10 of the *Ppie* gene (**d**). Each symbol represents a technical replicate (n = 3).Error bars represent mean ± s.d. **e** Fold change in the proportion of mCherry-positive cells 4 days after 4-OHT treatment in Type V (4716-10) cell line expressing sgRNAs targeting cyclophilin A and cyclophilin E. Empty Vector, β-Gal, and a p53 targeting sgRNAs used as controls. Each symbol represents a technical replicate (n = 3). Error bars represent mean ± s.d. Experiment was conducted twice, with similar results. **f** Flow cytometry-assisted cell viability assay in Type D (4711-18) cell line expressing sgRNAs targeting cyclophilin A, cyclophilin E, or both, 5 days after 4-OHT treatment. Live cell percentage determined by quantification of DAPI negative population. β-Gal, and a p53 targeting sgRNA used as controls. Each symbol represents a technical replicate (n = 3). Error bars represent mean ± s.d. Experiment was conducted in n = 2 independent Type D cell lines. DKO double-knockout. **g** Immunoblot analysis for PPIA, PPIE and p53 expression from whole cell extract (WCE) or immunoprecipitated FLAG-IP samples from HEK293T cells overexpressing FLAG-*Ppia* or FLAG-*Ppie*, and HA-*Trp53*. **h** Immunoblot analysis for PPIA, PPIE and p53 expression 24 h after CsA treatment from whole cell extracts (WCE) or immunoprecipitated (FLAG-IP) samples from HEK293T cells overexpressing FLAG-*Ppia* or FLAG-*Ppie*, and HA-*Trp53*. **i** Type D (4711-18) cells and Type V (4716-10) were treated with 4-OHT and CsA for 48 h and p53-Cyclophilin A (CypA) or p53-Cyclophilin E (CypE) interactions were assessed using proximity ligation assay. Representative insets are shown for each condition. Scale bars: 6 μm. **j** Quantification of the average number of PLA signals per cell shown in (**i**). Values are expressed as mean ± s.d. n = 3–6 random fields of view (>100 cells). Statistical significance was determined by one-way ANOVA followed by Tukey's multiple comparisons test. Source data are provided as a Source Data file.

metastatic mechanisms[10]. Thus, our findings warrant further investigation into the impact of cell-of-origin and driver mutations on p53-mediated tumor suppression in SCLC. These insights could identify context-dependent regulators of p53 function.

Cellular context influences the ability of p53 to induce specific tumor suppressive programs[36–39]. Moreover, p53 restoration in distinct cancer types was recently described to induce cancer type-specific p53 binding and transcriptional outputs that were likely responsible for differential induction of apoptosis in mouse lymphoma models, or senescence in sarcoma and lung adenocarcinoma models[40]. We profiled p53 binding and transcriptional output in Type D and Type V SCLC cells and observed that while global p53 binding was similar, p53 distinctly regulated a subset of genes between SCLC subtypes. Interestingly, although Module 5 genes are associated with diverse biological processes (e.g. vesicular trafficking, cell growth), they were enriched for some genes specifically involved with canonical forms of apoptosis (e.g. *Apaf1* and *Perp*) which may suggest some mechanistic overlap. However, the precise mechanism of p53-mediated cell death does not neatly fit into any defined cell death programs. That p53-mediated Type D SCLC cell death is abolished by cyclosporin A, and that cyclosporin A or CypA knockout similarly abolishes the induction of a distinct set of genes specifically induced in Type D SCLC, strongly suggests p53 orchestrates a deliberate program of cell death. However, the program is not associated with mitochondrial dysfunction, activation of caspases, or extensive fragmentation of DNA, and is not inhibited by small molecules targeting established determinants of major forms of programmed cell death. Elucidating the molecular underpinnings of p53-mediated Type D SCLC will be a major focus of future work that may identify novel therapeutic targets to activate this latent form of cell death in SCLC.

In summary, our study shows that p53 reactivation in SCLC identifies two tumor subtypes (Type V, Type D) that either undergo senescence or a cyclophilin-dependent cell death, and identifies cyclophilins as context-dependent regulators of p53-mediated transcription (Fig. 7).

## Methods

### Ethics statement
The research conducted in this study was conducted ethically and complies with all relevant guidelines and regulations. Animal studies were performed under strict compliance with Institutional Animal Care and Use Committee (IACUC) at University of Pennsylvania (#804774).

### Animal studies and treatment
*Kras*^LSL-G12D[57], *Trp53*^flox/flox[58], *Trp53*^XTR/XTR[21], *Rb1*^flox/flox, *p130*^flox/flox (*p130* is also known as *Rbl2*)[22] and *Rosa26*^FlpO-ER mice[24] have previously been described. Mice are mixed B6J/129S4vJae. Mice were transduced with $1.0 \times 10^8$ plaque-forming units (PFUs) of Ad:CMV-Cre obtained from University of Iowa Viral Vector Core[59]. Tumors were initiated before mice reached 20 weeks of age. For *Trp53* restoration, mice were treated with tamoxifen on two consecutive days with 200 μl of a 20 mg ml⁻¹ solution dissolved in 90% sterile corn oil and 10% ethanol by oral gavage. Weekly treatment with tamoxifen was administered for long-term experiments. For cyclophilin inhibition, mice were treated with Cyclosporine A with 200 μl of a 15 mg/kg solution dissolved in 90% sterile corn oil and 10% DMSO. The acquisition of micro-computed tomography (μCT) was performed on μCT setup (MI Labs) with Acquisition 11.0 software. Image reconstruction and visualization was performed with MI Labs REC-11.01. Individual and total tumor volumes were determined by constructing three-dimensional tomograms within ITK-SNAP v3.8 (*itksnap.org*)[60]. Mice in survival studies were monitored for lethargy, labored breathing and weight loss, at which time animals were euthanized. The size of each animal cohort was determined by estimating biologically relevant effect sizes between control and treated groups and then using the minimum number of animals that could reveal statistical significance using the indicated tests of significance. All animal studies were randomized in 'control' or 'treated' groups. However, all animals housed within the same cage were generally placed within the same treatment group. Animal sex was randomized at the time of treatment group assignment. Mice were housed in a vivarium with regulated light-dark cycles (12 h), ambient temperatures (20–24 °C), and relative humidity (45–65%).

### Allograft studies
SCLC cells were subcutaneously injected into the flanks of NCR^Nude mice (Taconic). Tumor volumes were estimated using Vernier calipers and animals were stratified into control or tamoxifen treatment groups when tumors reached a size of ~100 mm³. Tumor volumes were recorded bi-weekly through duration of experiment and never allowed to reach a combined tumor volume of 1 cm³.

### Immunohistochemistry and immunofluorescence
Lung and tumor tissues were dissected into 10% neutral-buffered formalin overnight at room temperature before dehydration in a graded alcohol series. Paraffin-embedded, H&E, and/or Trichrome stained histological sections were produced by the Penn Molecular Pathology and Imaging Core. Immunostaining for cleaved Caspase 3 (1:200,Cell Signaling 9661), ASCL1 (1:200, BD Biosciences 556604), UCHL1 (1:200, Sigma Aldrich HPA005993), CGRP (1:200, Sigma Aldrich C8198), F4/80 (1:500, Novus Biological NB600-404), GFP (1:1000, Abcam 13970), and

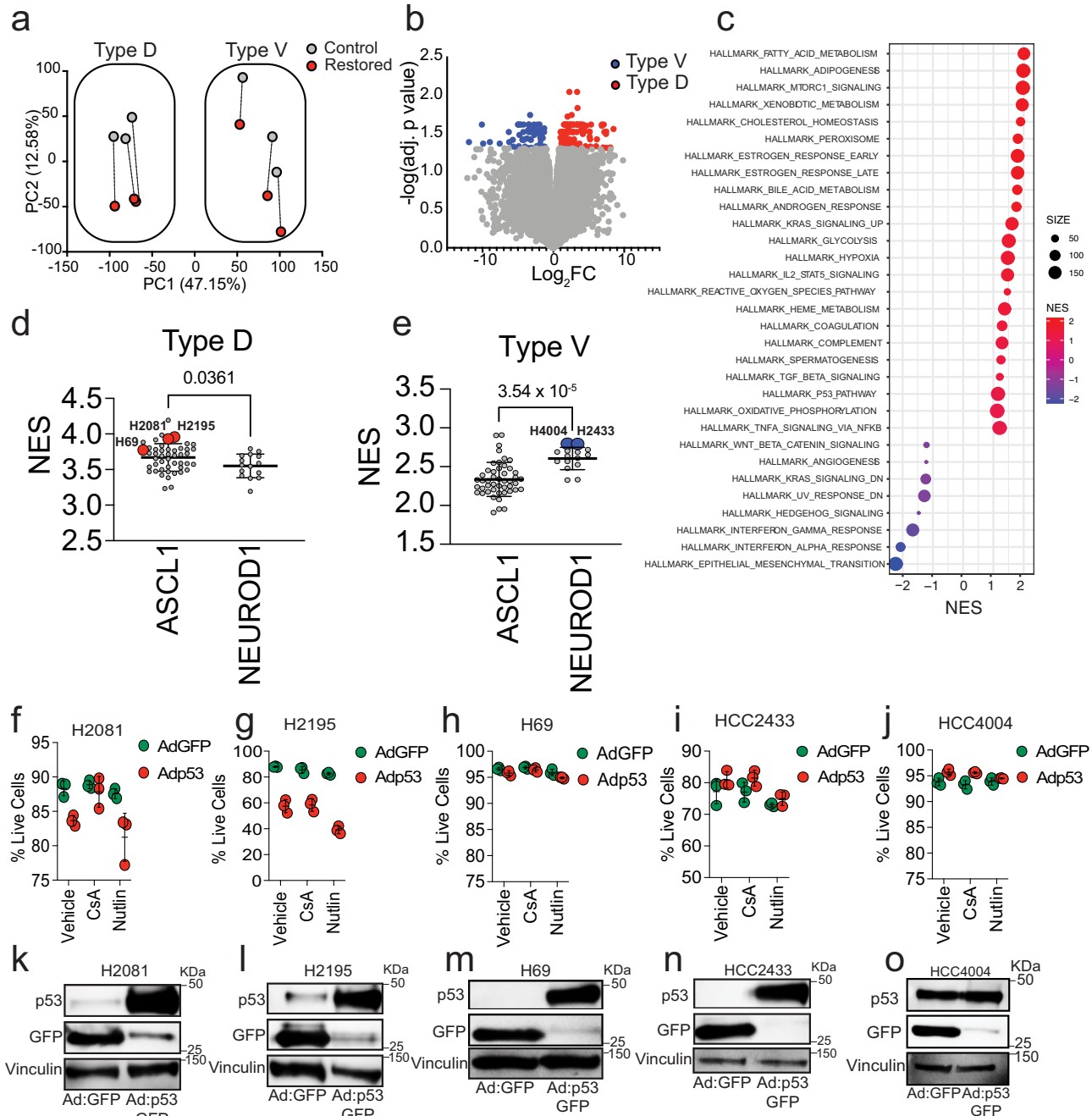

**Fig. 5 | Type V and Type D cells have features of distinct SCLC subtypes.**
**a** Principal component analysis of Type D (4711-1, 4711-18, 4716-11) and Type V (4711-11, 4716-10, 4716-14) cell lines 72 h after treatment with vehicle (Control) or 4-OHT (Restored). **b** Volcano plots of RNA-sequencing data in untreated Type D and Type V cells. Colored dots represent genes that are differentially enriched ($log_2$ fold-change greater-than 1 and false discovery rate (FDR)-adjusted $P$-value less-than 0.05) in Type D ($n = 109$ genes, red) or Type V ($n = 57$ genes, blue) cells. **c** Bubble plot representation of all GSEA hallmark gene sets with an FDR-adjusted $P$-value lower than 0.25 in untreated Type D (red) or Type V cells (blue). Normalized enrichment score (NES) plotted and the "SIZE" of each dot represents the number of genes within the gene set. **d**, **e** Dot plots indicating normalized enrichment scores of 'Type D' (**d**) and 'Type V' (**e**) gene signatures in human SCLC-ASCL1

($n = 47$) and SCLC-NEUROD1 ($n = 15$) cell lines. Single sample GSEA analysis was performed to quantify enrichment of specific gene signatures in gene expression datasets from human SCLC cell lines obtained using CellMiner-SCLC (Tlemsani et al. 2020). Error bars represent mean ± s.d. Statistical significance was determined by two-tailed Student's $t$-test. Highlighted cell lines were used in downstream analysis. **f**–**j** Flow cytometry-assisted cell viability assay in human SCLC (H2081, H2915, H69, HCC2433, HCC4004) cell lines infected with either Ad:GFP, Ad:p53-GFP, CsA and/or Nutlin for 48–72 h. Each symbol represents a technical replicate ($n = 3$). Error bars represent mean ± s.d. **k**–**o** Immunoblot analysis of p53 and GFP in human SCLC cell lines used in (**f**–**j**). Vinculin is loading control. Source data are provided as a Source Data file.

Ki67 (1:1000, Vector VP-RM04) were performed after citrate-based antigen retrieval. Cleaved Caspase 3, UCHL1, CGRP, F4/80, and ASCL1 staining was assessed by immunohistochemistry using ABC reagent (Vector Laboratories, PK-4001) and ImmPACT DAB (Vector

Laboratories, SK-4105) according to manufacturer instructions. GFP and Ki67 staining was assessed by immunofluorescence using a biotinylated anti-Rabbit secondary (Vector Laboratories, PK-4001), anti-chicken-Alexa594 (1:200, Jackson Immunoresearch), and a

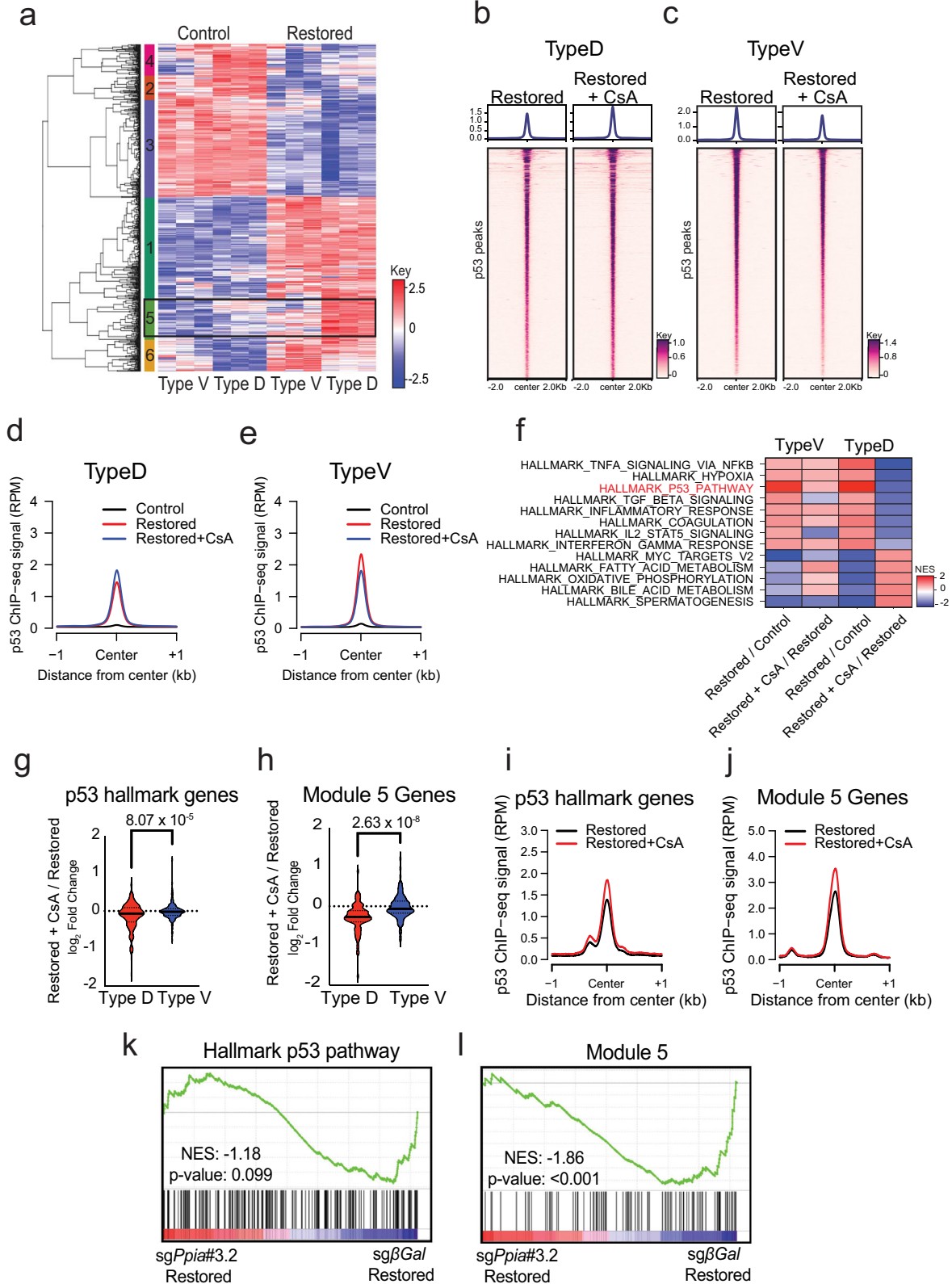

streptavidin-conjugated Alexa488 antibody (1:200, Thermo Fisher, S-32354).

Immunohistochemistry and immunofluorescence were both performed on paraffin-embedded sections following the same antigen-retrieval protocol. Sections were incubated in primary antibody overnight at 4 °C, secondary antibody for 1 h at room temperature, and for

immunofluorescence Streptavidin-conjugated fluorophore for 1 h at room temperature in the dark.

For TUNEL staining, tissues were deparaffinized and then permeabilized with 0.1% sodium citrate and 0.1% Triton-X in PBS for 8 min. TMR Red-conjugated TUNEL labeling mix (Millipore Sigma, 12156792910) was added to permeabilized tissue sections and incubated for 1 h at 37 °C in

**Fig. 6 | p53 controls a distinct transcriptional program in Type D SCLC that is dependent on cyclophilin activity. a** Heatmap representation of the union of differentially expressed genes identified in (Supplementary Fig. 16a, b) for Type D (4711-1, 4711-18, 4716-11) and Type V (4711-11, 4716-10, 4716-14) cells. Unbiased hierarchical clustering identifies 6 modules associated with distinct patterns of gene expression. The color key represents gene expression levels with darker colors representing higher (red) or lower (blue) gene expression. **b, c** Heatmap representation of ChIP-seq analysis for p53 bound peaks 48hrs after 4-OHT and CsA treatment in Type D (4711-1, 4711-18, 4716-11) (**b**) and Type V (**c**) (4711-11, 4716-10, 4716-14) cells. Heatmaps are centered on p53-bound peaks across a ± 2 kb window. **d, e** Metagene analysis of p53 ChIP-seq signal between Control, Restored and Restored + CsA in Type D ($n = 3$) (**d**) and Type V ($n = 3$) cells (**e**). Data are centered on p53-bound peaks across a ± 1 kb window. **f** Heatmap representation of GSEA hallmark gene sets with an FDR-adjusted *P*-value lower than 0.25 from the comparisons listed in the labels along the *y*-axis. The color key represents normalized enrichment score (NES). **g, h** Violin plots indicating $\log_2$ fold-change expression of canonical p53 targets from the GSEA gene set 'HALLMARK_P53_PATHWAY' (**g**) or Module 5 genes (**h**) in Restored and Restored + CsA Type D ($n = 3$) and Type V ($n = 3$) cells. Statistical significance determined by two-tailed Student's *t*-test. **i, j** Comparison of p53 ChIP-seq signal between Restored and Restored + CsA Type D ($n = 3$) cells for p53 hallmark genes (**i**) and Module 5 genes (**j**). Data are centered on p53-bound peaks across a ± 1 kb window. **k, l** GSEA of "HALLMARK_P53_PATH-WAY" (**k**) and Module 5 genes (**l**) in Type D (4711-1, 4711-18, 4716-11) cells expressing sgRNAs targeting β-Gal or *Ppia* 72 h after 4-OHT treatment. Nominal *p*-value calculated by permutation test using GSEA software. No adjustments were made since only one geneset was tested. Source data are provided as a Source Data file.

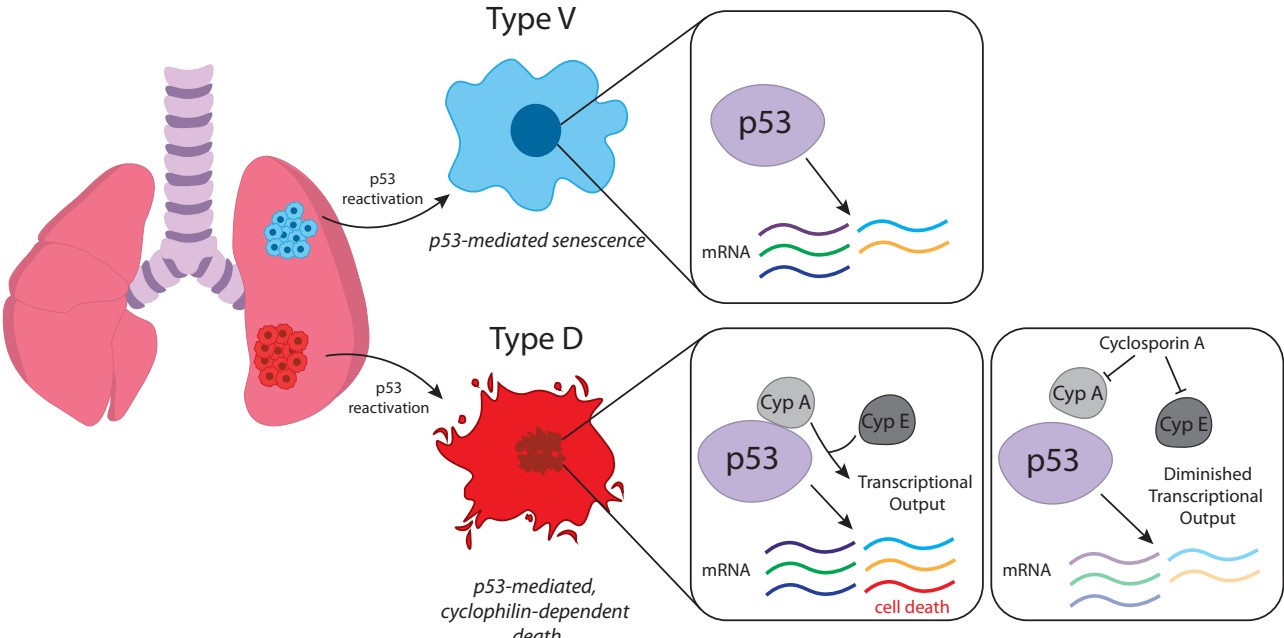

**Fig. 7 | Model: p53 induces cyclophilin-dependent cell death or senescence in distinct SCLC tumor subtypes.** Reactivation of p53 in small cell lung cancer identifies two tumor subtypes that are suppressed by induction of distinct p53-mediated tumor suppressive programs. Type V tumors undergo senescence after p53 reactivation whereas, Type D tumors undergo a form of p53-mediated, cyclophilin-dependent cell death. p53 controls the expression of widely similar genes between tumor subtypes; however, p53 reactivation in Type D cells specifically induces the expression of select genes (Module 5) that are associated with cell death but whose induction is dependent on the activity of cyclophilin. Cyclophilins may potentiate p53 transcriptional output in a direct (Cyclophilin A) or indirect manner (Cyclophilin E). Cyclosporine A (CsA) blocks p53-mediated death likely by abrogating CypA-p53 interactions and inhibition of CypE function, each of which are required for efficient induction of p53-mediated cell death in Type D small cell lung cancer.

the dark. For all immunofluorescence staining, nuclei were stained using 5 mg/ml DAPI at a 1:1000 dilution for 10 min, and then slides were mounted with Fluoro-Gel (EMS, 17985-50). All photomicrographs were captured on a Leica DMI6000B inverted light and fluorescence microscope, or a Leica M80 is a compact stereomicroscope.

### Senescence-associated beta galactosidase (SA-βGal) staining

SA-βGal staining conducted as previously described at pH 5.5 for mouse cells and tissue[61]. Frozen sections of lung tissue (14 μM) or adherent cells were fixed with 0.5% glutaraldehyde in PBS for 15 min washed with PBS supplemented with 1 mM $MgCl^2$ and stained for at least 8 h at 37 °C in PBS containing 1 mM $MgCl^2$, 1 mg/ml X-Gal, 5 mM potassium ferricyanide and 5 mM potassium ferrocyanide. Tissue sections were counterstained with eosin.

### Histological quantification

ImageJ software v1.51 was used to determine the frequency of cells that were positive for specified antigens (Ki67) as a fraction of total tumor cells. Data points represent individual tumors for Ki67 staining. For SA-βGal and TUNEL a binary staining score for each tumor was determined based on staining intensity scored from 0 (minimal staining) to 1 (ubiquitous staining). For Trichrome staining, a similar staining scoring system was applied where individual tumors were scored from 0 (minimal necrosis or fibrosis), to 1 (ubiquitous necrosis or fibrosis). Data points represent either individual animals or tumors, as indicated.

### Cell lines

Primary mouse tumors were mechanically separated using scissors under sterile conditions and cultured in high-glucose DMEM supplemented with 10% fetal bovine serum, GlutaMAX and antibiotics at 37 °C and 5% $CO_2$ until cell line establishment. Cell lines were authenticated for genotype. The $p53^{XTR/XTR}$ allele was validated via three-primer PCR reactions using purified genomic DNA as a template to detect *XTR*, *TR*, *R* and *WT* alleles. Primers used were as follows: (1) 5′- cttggagaca-tagccacactg -3′, (2) 5′- caactgttctacctcaagagcc −3′ and (3) 5′- cttgaa-gaagatggtgcg −3′. Cell lines were further validated for the expected,

genotype-associated protein expression patterns by western blot. KP^Restorable cell lines used in senescence experiments were derived from *Kras*^LA2/+;*Trp53*^LSL/LSL;*Rosa26*^CreERT2/CreERT2 adenocarcinomas as previously described (Feldser, et al. 2010). All cell lines were tested for mycoplasma using MycoAlert detection kits (Lonza) as per manufacturer instructions. Cell lines are available upon request.

Cell number was counted using a Beckman Coulter Z2 Cell and Particle Counter. 4-Hydroxytamoxifen (4-OHT) dissolved in ethanol was administered once at the time of cell plating at a final concentration of 500 nM. AdCre or AdFlpO was purchased from the University of Iowa Gene Vector Core and administered to adherent cells at time of cell plating. Proliferation of cell lines was determined by plating the indicated cell number and analysed by Coulter Counter cell counts at the indicated time points. Photomicrographs were captured on a Leica DM IL LED inverted light and fluorescence microscope. For live cell imaging, photomicrographs were obtained in a 37 °C humidified chamber every 15 min for 4 days using a Leica DMI6000B microscope.

HEK293FT cells that were used for lentivirus production were obtained from Invitrogen. HEK293FT cells were cultured in DMEM containing 10% Bovine Serum.

HEK293FT cells were validated by verifying that high-titer virus production was possible. Lentivirus production was performed as described previously[62].

Human SCLC cell lines were acquired from ATCC (H69, H2081, H2195) or UT Southwestern's SCLC cell line repository (HCC2433, HCC4004). H69, HCC2433, and HCC4004 were cultured in RPMI supplemented with 5–10% FBS, H2081 and H2195 were cultured in DMEM/F12 supplemented with 5% FBS and HITES (10 nM hydrocortisone, Insulin-Transferrin-Selenium (Gibco), and 10 nM beta-estradiol). For wild-type p53 reintroduction, AdGFP and Adp53-GFP were purchased from the University of Iowa Gene Vector Core or Vector Biolabs and administered to the cells at time of cell plating.

## Immunoblot analysis

Cells were lysed in RIPA buffer, resolved on NuPage 4–12% Bis-Tris protein gels (Thermo Fisher) and transferred to polyvinylidene fluoride (PVDF) membranes. Blocking, primary and secondary antibody incubations were performed in Tris-buffered saline (TBS) with 0.1% Tween-20. Atg5 (1:5000, Cell Signaling, 12994 S) Bcl2 (1:2000, Abcam, ab182858), CDKN2A/p19ARF (1:200, Santa Cruz, sc-32748), GFP (1:1000, Cell Signaling Technology, 2956 S) Human p53 (1:200, Santa Cruz, sc-126), mouse p53 (1:2500, Leica Biosystems, P53-PROTEIN-CM5), p21 (1:500, Santa Cruz, sc-6246), Cleaved Caspase-8 (1:1000,Cell Signaling Technology, 8592 S), Cleaved Caspase-3 (1:1000,Cell Signaling Technology, 9661 S), Cleaved PARP (1:1000,Cell Signaling Technology, 9548 S), RB (1:1000, Abcam, ab181616), ASCL1 (1:1000, BD Biosciences 556604), PPIE (Cyclophilin E, 1:500, Novus, H00010450-M02), PPIF (Cyclophilin D, 1:1000, Abcam, ab110324), PPIA (Cyclophilin A, 1:1000, Cell Signaling Technology, 2175 S), β-actin (1:10000,Sigma Aldrich, A2066), HSP90 (1:10000, BD Transduction Laboratories, 610418), GAPDH (1:10000,Cell Signaling Technology, 2118 S), COXV (1:4000,Cell Signaling Technology, 11967 S), H3 (1:10000, Abcam ab1791), Vinculin (1:8000, Santa Cruz, sc-73614) were assessed by western blotting. HSP90, GAPDH, β-actin, COXIV, Vinculin, and H3 were used as loading controls. Protein concentration was determined using a BCA protein assay kit (Pierce).

## Flow cytometry

For in vitro experiments, single cell suspensions were prepared from SCLC cells after collection by passing specimens through a 100-μM cell strainer. Cells were resuspended in FACS buffer. Equal cell numbers of cells were plated and stained for flow cytometry analysis. Cell cycle analysis was performed according to the manufacturer's instructions for the APC BrdU Flow Kit (BD Pharmingen). TUNEL staining was performed according to the manufacturer's instructions for the In Situ

Cell Death Detection Kit, TMR red (Millipore Sigma, 12156792910). For cell viability assays, cells were resuspended in FACS buffer containing DAPI at a 1:1000 dilution. For all in vitro experiments, flow cytometry was performed using an Attune NxT flow cytometer (Thermo Fisher). Data were analyzed using FlowJo v10.8. Data points represent replicates, or means from independent experiments, as indicated.

For in vivo experiments, tumors were microdissected directly from the lungs of *RP*^X*R2* mice and individually placed in 500 μl of tumor digestion buffer consisting of PBS containing 10 mM HEPES pH 7.4, 150 mM NaCl, 5 mM KCl, 1 mM MgCl₂, and 1.8 mM CaCl₂, along with freshly added Collagenase 4 (Worthington 100 mg/ml solution, 20 μl per ml of digestion buffer) and DNase I (Roche 10 mg/ml solution, 4 μl per ml of digestion buffer). Tumors were manually dissociated using scissors, and then placed in a 4 °C shaker for 1 h at 250 rpm. Digested tumors were then filtered into strainer-cap flow tubes (Corning, 352235) containing 1 ml of horse serum (Thermo Fisher, 16050122) to quench the digestion reaction. Cells were spun down at 200 g for 5 min with the cap in place to obtain all cells. The supernatant was aspirated, cells were washed once with PBS and resuspended in FACS buffer. Cells were labeled with the following antibodies: PD-1 FITC (1:50, Biolegend), NKp46-PE (1:50, Biolegend), CD103-PETR (1:100, Biolegend), CD3-PE/Cy5 (1:100, Biolegend), CD8-PE/Cy7 (1:100, Biolegend), CD44-APC (1:100, Biolegend), CD45-AF700 (1:400, Biolegend), F4/80-APC/Cy7 (1:100, Biolegend), CD11b-PerCP/Cy5.5 (1:200, BD Biosciences), CD11c-QD605 (1:100, Biolegend), CD45-BV570 (1:100, Biolegend), CD4-BV650 (1:50, Biolegend), Live/Dead Aqua (1:600, Invitrogen). For all in vivo experiments, flow cytometry was performed using a BD LSR II flow cytometer (BD Biosciences). Data were analyzed using FlowJo v10.8. Data points represent individual tumors.

## Drugs, cell death inhibitors

Drugs or cell death inhibitors are used at the following concentrations: Necrostatin-1s (10 μg/mL, Selleck Chemicals), Ferrostatin-1 (2 μM, XcessBio), Liproxstatin-1 (2 μM, Sellek Chemicals), Vitamin E (125 μM, Sigma Aldrich), Cyclosporin A (1 μM, Sigma Aldrich), ZVAD-FMK (20 μM, Abcam), NIM-811 (1 μM, MedChem Express), FK506 (1 μM, ApexBio), Nutlin-3a (10 μM, ApexBio). Cell death inhibitors were added to cells at time of plating for the duration of the experiment.

## CRISPR design and production

Validated sgRNAs targeting *Atg5* were obtained from Genscript. All remaining sgRNAs used were designed using the Sanjana Lab CRISPR Cas9 library design tool (http://guides.sanjanalab.org), or Benchling, using the CRISPR Design Tool. sgRNAs targeting *βgal* and *Trp53* were used as experimental controls[63]. All sgRNA duplexes were Golden Gate-cloned into BsmBI sites of the LentiCRISPRv2-Puro or LentiCRISPRv2-mCherry vectors[62,64,65]. The sgRNA sequences used for targeting Cas9 are detailed in Supplementary Table 1. CRISPR vectors used in this study are available upon request.

## Transfection-mediated gene transfer

HEK293T cells were transfected with plasmids using polyethylenimine (PEI) (Polysciences, #24765). 4.0 μg total plasmid DNA was transfected for 48 h.

## Immunoprecipitation and immunoblot

Cells were lysed in NP-40 buffer (0.1% NP-40, 15 mM Tris−HCl pH7.4, 1 mM EDTA, 150 mM NaCl, 1 mM MgCl2, 10% Glycerol) containing protease inhibitors (Sigma, #11697498001) and the lysates were incubated with anti-FLAG M2 Affinity Gel (Sigma, #A2220) at 4 °C for 2 h. After five washes with NP-40 buffer, the anti-FLAG Gel was mixed with Laemmli buffer and boiled at 95 °C for 5 min. After SDS−PAGE electrophoresis and transfer, primary antibodies and HRP-linked secondary antibodies were incubated with the membrane for 2 h at room temperature and overnight at 4 °C, respectively. After washing with

PBS-T three times and PBS once, the membrane was detected by the chemiluminescence system (Thermo Fisher Scientific, #32106). Where indicated, Cyclosporin-A was included during lysis, anti-FLAG Gel incubation, and subsequent washes. The following antibodies were used: anti-FLAG (Sigma, #F7425, 1:6,000), anti-HA (Cell Signaling Technology, #3724, 1:2,000), and anti-Rabbit IgG-HRP (GE Healthcare, #NA934V, 1:10,000).

## Proximity ligation assays

Protein-protein interactions were assessed using the PLA Duolink in situ starter kit (Sigma-Aldrich) following the manufacturer's protocol. In brief, cells were grown on eight-well chamber slides (Lab-Tek), and were left untreated or treated with 4-OHT and/or CsA for 48 h. Cells were fixed in 4% paraformaldehyde (Electron Microscopy Sciences) for 10 min, followed by 3 washes in 1X PBS and permeabilization with 0.25% Triton X-100 (Millipore Sigma) for 5 min. Cells were incubated in blocking solution (Sigma-Aldrich) in a humidified chamber for 1 h at 37 °C, followed by incubation in a humidified chamber for 1 h at 37 °C with the following primary antibodies: p53 (1:1000, 1C12, Cell Signaling #2524), p53 (1:200,Leica Biosystems, P53-PROTEIN-CM5), Cyclophilin A (1:250,Thermo, #PA1-025), Cyclophilin D (5 ug/mL, Abcam, ab110324), Cyclophilin E (10 ug/mL, Novus, H00010450-M02), Tomm20 (1:400, D8T4N, Cell Signaling #42406). Cells were then incubated with PLA Probes (Sigma-Aldrich) 1 h at 37 °C, followed by ligation (30 min) and amplification (100 min) at 37 °C. Cells were mounted with media containing DAPI and images were captured using a Leica TSC SP5 confocal microscope.

## Plasmids

pcDNA3.1 + C-HA Trp53 (OMu22847) and pcDNA3.1 + C-FLAG *Ppia* (OMu14516), pcDNA3.1 + C-FLAG *Ppie* (OMu11106) plasmids used in mammalian over-expression experiments were purchased from Genscript. pMSCV-*Bcl2*-IRES mCherry plasmid used in mammalian over-expression experiments in Type D cells was generated by cloning *Bcl2* cDNA PCR amplified from tetO-Bcl2-IRES-tdtomato (Addgene Plasmid #117857) into pMSCV-IRES-mCherry FP (Addgene Plasmid #52114) using NEBuilder HiFi DNA Assembly (New England Biolabs #E2621S). Plasmids used in this study are available upon request.

## Seahorse XF cell mito stress analysis

Oxidative respiration was measured using XF Cell Mito Stress Test Kit (Agilent Technologies, 103015-100). $1 \times 10^4$ cells per well were seeded on an XF96 Cell Culture Microplate. Each cell line was split into two groups: control (EtOH) or restored (4-OHT) and 6 replicates were plated. Microplate was incubated for 24 h at 37 C. Seahorse XF96 FluxPak sensor cartridge was hydrated with 200 µl of Seahorse Calibrant in a non-CO2 incubator at 37 C overnight. After 24 h, cells were incubated with base medium (Agilent Technologies, 102353-100) containing 2 mM L-glutamine, 1 mM sodium pyruvate, and 10 mM glucose in a non-CO2 incubator at 37 C for 45 min prior to assay. Oxygen consumption rate (OCR) was measured by XFe96 extracellular flux analyzer with sequential injections of 1 µM oligomycin, 1 µM FCCP, and 0.5 µM rotenone/antimycin A. After the run, cells were lysed with 15 µl RIPA buffer and protein concentration was quantified using BCA Protein Assay Kit (Thermo Scientific). OCR measurements were normalized to the protein concentration in each well.

## Cellular protein fractionation

For mitochondrial fractionations, cells were processed using Pierce Mitochondria Isolation Kit for Cultured Cells (Thermo Scientific) as per the manufacturer's instructions.

## RNA isolation and RNA-sequencing

3 Type D and 3 Type V SCLC tumor-derived cell lines were treated with either vehicle (EtOH), 4-OHT, and/or CsA for 3 days and RNA was isolated using the RNeasy Mini Kit (Qiagen) as per the manufacturer's instructions. RNA concentrations were measured using the Qubit RNA Assay Kit (Invitrogen) and sample quality was determined using the RNA 6000 Nano Kit on a 2100 BioAnalyzer (Agilent). Sequencing libraries were prepared using the TruSeq Stranded mRNA Library Prep Kit (Illumina) as per the manufacturer's instructions and library quality was determined used the Agilent DNA 1000 Kit on a 2100 BioAnalyzer. Validated libraries were subjected to 75-bp single-end sequencing on the Illumina NextSeq 500 platform. For RNA-sequencing of CRISPR-targeted Type D cells (*BGal,Ppia*), 3 cell lines per sgRNA were treated with either vehicle or 4-OHT for 3 days. RNA was isolated using the RNeasy Mini Kit (Qiagen) as per the manufacturer's instructions and samples were submitted to Azenta Life Sciences for quality control, library preparations, sequencing reactions and initial bioinformatics analysis.

Fastq files for each sample were aligned against the mouse genome mm10 (GRCm39) using the Kallisto pseudoaligner v0.46.2, or HISAT2 v2.2.1. Differential expression analysis was carried out with the Limma tool v3.44.3 in R, or DeSeq2 v1.38.3. For volcano plot data, differentially enriched genes were defined as genes with a false-discovery rate adjusted P value less than 0.05 and $\log_2$-normalized fold change in expression greater than $\pm 1$. Geneset enrichment analysis was done using the GSEABase v1.60.0 and msigdbr v7.5.1 packages in R, or using the GSEA v4.2.2 software provided by the Broad institute (https://www.gsea-msigdb.org/gsea/index.jsp). 'Type D' and 'Type V' gene signatures were defined as genes with a false-discovery rate adjusted P value less than 0.05 and $\log_2$-normalized fold change in expression greater than 1 (109 genes were differentially regulated in Type D cells, 57 genes were differentially regulated in Type V cells). Human SCLC datasets for SCLC-ASCL1 and SCLC-NEUROD1 signature enrichment were obtained from CellMiner-SCLC (https://discover.nci.nih.gov/SclcCellMinerCDB/)[44]. Single sample GSEA was performed using the GSVA package v1.46.0 in R. Gene ontology analysis was performed using the gProfiler2 package v0.2.1 in R, or the g:Profiler web tool (https://biit.cs.ut.ee/gprofiler/gost). Visualization of gene expression patterns (heatmaps) and GSEA data was performed using the gplots package v3.1.3 in R. Other visualization and statistical analysis were performed using Prism 9.

## Chromatin immunoprecipitation (ChIP)

3 Type D and 3 Type V SCLC tumor-derived cell lines were treated with either vehicle (EtOH) or 4-OHT for 2 days for ChIP. $1 \times 10^7$ SCLC cells were harvested and cross-linked at room temperature by resuspending in PBS containing 1% formaldehyde for 2.5 min. The reaction was quenched by addition of glycine to a final concentration of 0.125 M. After washing with cold 1X PBS, cells were lysed in 1 mL sonication buffer (0.25% Sarkosyl, 1 mM DTT, and protease inhibitors into RIPA buffer). Cell suspension was sonicated using a Covaris S220 Focused Ultrasonicator until chromatin was sheared to a size range of around 200 bp. Lysates were centrifuged $18k \times g$ for 5 min and supernatant was collected. After saving 10% for an input sample, 20 uL of anti-p53 antibody (CM5, Leica Biosystems was added to lysate and incubated overnight. 200uL of a bead slurry prepared from Protein A/G Magnetic Beads (Thermo Scientific) were added to lysates and incubated in a rotator at 4 °C for 4 h. Chromatin-bound beads were washed three times with low-salt wash buffer (0.1% SDS, 1% Triton X-100, 1 mM EDTA pH 8.0, 50 mM Tris-HCl at pH 8.0, 150 mM NaCl), high-salt wash buffer (0.1% SDS, 1% Triton X-100, 1 mM EDTA pH 8.0, 50 mM Tris-HCl pH 8.0, 500 mM NaCl), and LiCl wash buffer (150 mM LiCl, 0.1% SDS, 0.5% deoxycholic acid sodium salt, 1% NP-40, 1 mM EDTA pH 8.0, 50 mM Tris pH 8.0). After washing the beads once with TE buffer containing 50 mM NaCl, the beads were resuspended in ChIP elution buffer (1% SDS, 200 mM NaCl, 10 mM EDTA pH 8.0, 50 mM Tris pH 8.0) and eluted for 30 min at 65 °C. Bead solution was spun down for 1 min at 16 g and 200uL of the supernatant were transferred to a new tube. 10%

input was diluted to 200uL using ChIP elution buffer. Diluted input and experimental samples were reverse cross-linked overnight at 65 °C. All samples were treated with RNAse A and incubated at 37 °C for an hour, followed by Proteinase K treatment and incubated for an hour at 55 °C. DNA was recovered by PCR Purification Kit (QIAGEN) and eluted into 50uL of molecular biology grade water. Chromatin-immunoprecipated DNA was validated for p53 target enrichment by qPCR using SYBR Green Power Up (Thermo Fisher) and ViiA7 Real-Time PCR System (Thermo Fisher). Sequencing libraries were prepared using the NEBNext Ultra II DNA Library Prep kit for Illumina (New England Biolabs) size selection, as per the manufacturer's instructions, with the following conditions: 30 uL ChIP sample or 5 uL input sample used when end prepping, adaptor was diluted 10-fold for ChIP samples but not diluted for input, 200 bp bead based size selection used for all samples, 14 cycles used for PCR enrichment of ChIP samples and 7 cycles for input samples. Library quality was determined used the Agilent DNA 1000 Kit on a 2100 BioAnalyzer (Agilent). Validated libraries were subjected to 75-bp single-end sequencing on the Illumina NextSeq 500 platform.

### ChIP-sequencing

p53 ChIP-seq data was aligned to mouse genome mm10 (GRCm39) by bowtie2 v2.4.5 with default parameters. Peak calling was performed by MACS2 with a *p*-value threshold of 1E-10. ChIP-seq signal in a specific region was calculated by the normalized RPM. Briefly, ChIP-seq reads aligning to the region were extended by 100 bp and the density of reads per bp was calculated using featureCounts v2.0.2. The density of reads in each region was normalized to the total number of million mapped reads, producing read density in units of reads per million mapped reads per bp (RPM per bp). Differential p53 binding analysis between Type D and Type V cells was carried out using the DiffBind v3.1 tool in R. Differential regions were identified with fold change and *p*-value cutoffs, as indicated. The nearest gene of specific p53 peak was treated as the associated gene of p53, which was identified by the HOMER v4.11 module annotatePeaks.pl.

### Statistics and reproducibility

All analyses were performed in the Graphpad Prism 9 software package. For survival studies, log-rank (Mantel–Cox) tests were performed to determine significance. For SA-βGal contingency analysis, Fisher's exact test was performed. For cyclophilin and ferroptosis inhibition experiments, one-way ANOVA followed by Dunnett's multiple comparison tests were performed. For proximity ligation assay experiments, one-way ANOVA followed by Tukey's multiple comparison tests were performed. For all remaining experiments, unpaired Student's *t*-tests were performed to determine significance, as indicated. No statistical method was used to predetermine sample size. No data were excluded from the analyses. Allocation of mice into experimental groups was randomized. Other experiments were not randomized. Researchers were not blinded to sample identity and group except for histopathological assessments of necrosis, fibrosis, and senescence. For Fig. 1c, f, the experiments were conducted once on at least 3 separate mice per experimental group, with similar results. For Fig. 1h, the experiment was conducted once on 3 separate mice, with similar results. For Fig. 3d, the experiment was conducted at least 2 times on independent cell lines per SCLC subtype, with similar results. For Fig. 3h, the experiment was conducted 2 times on 3 independent cell lines, with similar results. For Fig. 3i, the experiment was conducted once on 2 independent cell lines per SCLC subtype, with similar results. For Fig. 3j, the experiment was conducted once on 3 independent cell lines, with similar results. For Fig. 3k, the experiment was conducted once on 2 independent Type D cell lines, with similar results. For Fig. 3l, the experiment was conducted three times, with similar results. For Fig. 4g, h, the experiments were conducted once.

For Figs. 5k–o, the experiments were conducted once. For Supplementary Fig. 1, experiment was conducted once. For Supplementary Fig. 2a, experiment was conducted once in 8 *RP^{TR}R2* independent cell lines, with similar results. For Supplementary Fig. 2b, experiment was conducted once with at least two independent cell lines per experimental group, with similar results. For Supplementary Fig. 2e, experiment was conducted once in 8 *RP^{TR}R2* independent cell lines, with similar results. For Supplementary Fig. 4e, the experiment was conducted once on 2 independent cell lines per SCLC subtype, with similar results. For Supplementary Fig. 4f, the experiment was conducted once on 3 independent cell lines, with similar results. For Supplementary Fig. 5d, experiment was conducted once on 2 independent cell lines, with similar results. For Supplementary Fig. 6d, experiment was conducted two times, with similar results. For Supplementary Fig. 10d, the experiment was conducted once on 3 independent cell lines, with similar results. For Supplementary Fig. 11a, experiment was conducted once. For Supplementary Fig. 11b, experiment was conducted once on 3 independent cell lines, with similar results. For Supplementary Fig. 12f, experiment was conducted once on 2 independent Type D cell lines, with similar results. For Supplementary Fig. 13a, experiment was conducted once. For Supplementary Fig. 14a, experiment was conducted once on 3 independent cell lines per SCLC subtype, with similar results. For Supplementary Fig. 14b, the experiment was conducted once on 2 independent cell lines per SCLC subtype, with similar results. For Supplementary Fig. 14c, experiment was conducted once. For Supplementary Fig. 18a, experiment was conducted once.

### Reporting summary

Further information on research design is available in the Nature Portfolio Reporting Summary linked to this article.

## Data availability

RNA and ChIP sequencing data associated with this study have been deposited in the Gene Expression Omnibus (GEO) under accession number: GSE231705. There are no restrictions on these data. Source data are provided with this paper.

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

## Acknowledgements

We would like to thank ULAR staff for animal husbandry, the Molecular Pathology and Imaging Core for histological analysis and technical expertise, E. Blankemeyer for computed tomography, D. MacPherson for *RP* cell line, I. Asangani and members of his lab for help with Next Generation Sequencing, J. Minna and V. Stastny for help with human SCLC cell line studies, and M. Winslow and members of the Feldser Laboratory for manuscript critique. This work is supported by: NIH grants (R01CA222503, and R21CA205340 to D.M.F., and F31CA254405 to J.A.), and the U.S. Department of Defense (LC210440 to D.M.F.).

## Author contributions

J.A., M.C., and K.M.A. performed animal studies. J.A. performed micro-CT analyses. J.A., A.I., N.M., H.M., M.C., and M.R.T., performed cell culture studies. A.C.G. performed Seahorse Mito Stress analysis. K.M.A. and J.L. performed immunological analysis with supervision from B.Z.S. J.A., Q.L., and N.F.F. performed bioinformatics analyses. L.W. supervised ChIP and bioinformatic analysis. G.P.G. conducted co-immunoprecipitation analysis with supervision from L.B. A.I. performed proximity ligation assay experiments and data analysis with supervision from M.M. J.A., M.C., K.R.D., and K.L.M. performed histopathological analyses and quantification. J.A., and D.M.F. interpreted all datasets. D.M.F. and J.A. conceived and designed the project, and wrote the manuscript with editorial help from A.C.G.

## Competing interests

The authors declare no competing interests.
