## [Peer Review File · Nature Communications]

p53 restoration in small cell lung cancer identifies a latent Cyclophilin-dependent necrosis mechanismREVIEWER COMMENTS

Reviewer #1 (Remarks to the Author); expert in p53, senescence and cell death:

The manuscript "p53 restoration in small cell lung cancer identifies a latent cyclophilin-dependent necrosis mechanism" leverages a model that allows conditional restoration of WT trp53 in a small cell lung cancer GEMM to demonstrate a novel connection between cell lineage and p53 response. Upon p53 restoration SCLC tumors undergo either a senescence like arrest or regression. These regressions are explained by cell culture-based studies demonstrating that p53 triggers a non-apoptotic, cyclophilin dependent necrosis in distinct lineages. This is a provocative manuscript which identifies an unappreciated mediator of p53 mediated cell fate.

The strength of this work lies in the innovative genetic model and clever use of inhibitors to identify the nature of the observed cell death. However, there are several important control and validation experiments missing that are required to fill in the gap between pharmacological and genetic approaches to support the proposed mechanism as well as rule out alternative explanations for the observed effects.

Major points:

1. While the cyclosporin A rescue of cell death in vitro upon p53 restoration provides a reasonably convincing starting point for a cyclophilin dependent death mechanism, the genetic rescue by sgRNA deletion of cyclophilin A and E lacks several important controls that will help explain variable and/or phenotypically weak results. In the initial sgRNA experiment and validation in different cell lines (Fig3E-G) the effect of targeting Ppia provides either dubious rescue (3F) or a highly variable one (3G). Is this because of variable editing efficiency or cell line specific/contextual roles for the different cyclophilins? The authors should show western blots and/or sequencing evidence demonstrating the level of knockdown or editing efficiency corresponding to the degree to which competitive fitness or viability is rescued. Given the ability of one of the Ppie sgRNAs to rescue to a similar extent as sg-p53 in 3G, the authors should generate a series of single and double Ppia and Ppie knockout "D" lines with accompanying analysis of both knockout efficiency and viability. This will provide a more specific mechanism and elucidate a potential hierarchy between Ppia and Ppie. This is especially salient since p53 appears to bind to Ppia and not Ppie in these cells, raising the question of the molecular "order of events" in this process that are dependent on cyclophilin binding versus binding independent functions. Another important control that I could not find is the effect of cyclophilin knockout in "V" type cells. Presumably their knockout would not alter the trajectory upon p53 restoration in these cells.
2. Furthermore, although baseline cyclophilin gene expression (Fig S8) does not explain the "V" versus "D" phenotype, it is not clear if different cyclophilin protein levels and distinct patterns of cyclophilin binding can explain the distinct p53-dependent biology in the 2 classes of cells. Unless I missed it, the binding experiments were all performed in an overexpression setting in 293T cells rather than with endogenous proteins in the primary cell lines. The protein levels of cyclophilins in the different classes of cell lines as well as the ability of endogenous p53 and cyclophilins to bind should be tested.
3. The authors propose that p53-cyclophilin binding directs the "D" versus "V" phenotype due to the ability of cyclosporin A to blunt "D" specific genes upon p53 restoration. 2 important experiments are missing to properly interpret this. First, p53 accumulation must be measured in the setting of cyclosporin A treatment and in genetic experiments with specific cyclophilin knockdown. Can cyclophilin binding alter p53 stability? Both p53 hallmark genes and the cluster 5 genes are reduced in restored cells with cyclosporin A treatment, along with a modest reduction in ChIP signal. This would be consistent with reduced p53 levels in cells. Also, as mentioned in point 1, the same sgRNA system with validated knockout/editing efficiency should be used to validate cyclophilin dependency of these gene expression changes. RNA SEQ is not necessary, but a set of presumably cyclophilin dependent genes should be assayed up p53 restoration in sgPpia, sgPpie, and double sgPpia/Ppie cells. Again, this would demonstrate the cyclophilin specificity of this phenotype.

4. In vivo rescue of regressions by cyclosporin A is a compelling result, but could be due to non-specific effects of the drug on tumor cells and the microenvironment. The authors should perform the allograft p53 restoration experiments shown in Figure 3K with a set of "D" cell lines expressing sgRNAs targeting Ppia, Ppie, and both sgPpia/Ppie. This will address if the cell death phenotype in vivo is indeed cyclophilin specific and cell intrinsic.

Minor points

1. The author's use of a spectrum of well-established inhibitors of cell death is a convincing approach to identify specific mechanisms by which "D" cells die. They state that while p53 restoration results in cyclosporin A sensitive necrosis, it is not characterized by notable post-translation activity of p53 at the mitochondria (e.g. Vaseva et al., 2012). However, there does appear to be more p53 in the mitochondrial fraction of the "D" versus "V" cells shown in S5I. Is this observed consistently across additional "D" and "V" cell lines? To better rule out the mPTP route the authors should gauge another functional characteristic of that path to cell death, such as mitochondrial swelling, or perform an orthogonal, more sensitive, test for p53 localization (e.g. co-staining of p53 and mitochondria).

2. IHC for p53 and representative direct targets should be performed in tissues after p53 restoration. While tamoxifen treatment clearly alters the growth of tumors, the extent of p53 accumulation is associated with restoration associated phenotypes in vivo (notably in mutant Kras driven lung cancers). Is heterogeneity in vivo also due differences in p53 accumulation or activity as measured by a target like p21?

3. In Figure 5, the authors should label the matched control versus restored pairs of "D" and "V" cells. It is unclear if p53 restoration in the "V" cells has a coherent effect in PC2 as it does in the "D" cells.

4. There are several typos and fragmented sentences throughout.

Reviewer #2 (Remarks to the Author); expert in lung cancer and genetics:

Thank you for the opportunity to review this manuscript. In this work, the authors utilize a conditional knockout/knock-in murine model to demonstrate the requirement for persistent p53 inactivation in sustaining established SCLC. They showed that tamoxifen-mediated p53 reactivation in mice with established tumors prolonged survival and induced two pathways for cell death in SCLC—a canonical senescence response (Type V) and necrotic cell death pathway (Type D) that is molecularly distinct from known cell death pathways and dependent on cyclophilin A and E. Treatment with CsA reduced cell death caused by restoration of p53.

Interestingly, they showed that Type D and Type V cells were enriched for recently described human SCLC subtype specific gene signatures—with type D being similar to the classic SCLC-A and type V were enriched for the SCLC-N gene signature.

Overall, this study was well executed and the data and discussion are clear. There are no major methodologic concerns, though the data are intrinsically limited by the efficacy of p53 restoration.

The major weaknesses include lack of clear relevance to SCLC therapy, as they predict that CsA would block these therapies, no strategy to enhance efficacy of restoration is proposed, and no effective p53 restoration therapeutics are on the horizon. Were there any clues from the analysis of Type V cells that suggests specific therapeutic vulnerabilities? Also notably, p53 restoration in human SCLC-A and SCLC-N cell lines was not shown to corroborate these findings. Figure 2D: Ki67+ of ~20% seems very low for human SCLC.

Reviewer #3 (Remarks to the Author); expert in p53 and cell death:

The absence of the tumour suppressor p53 drives tumour development. Previous elegant work, some of it from the authors of this manuscript, showed that restoration of wild-type p53 expression can substantially inhibit the expansion of tumours that were driven by loss of p53 (in combination with expression of certain oncogenes or loss of additional tumour suppressors). In lymphoid malignancies, this inhibition of tumour expansion is due to wild-type p53 induced apoptosis of the malignant cells. In liver and lung cancer models this was shown to be due to wild-type p53 induced senescence of the malignant cells.

In this manuscript, Jonuelle Acosta and colleagues present results from experiments using their innovative new model of lung cancer development, driven by loss of p53 combined with loss of RB, in which they can restore expression of wild-type p53 in the malignant cells (in vivo or in vitro). They show that the lung cancers that ostensibly have similar phenotypes (similar expression of marker proteins) fall into two groups in regard to their response to restoration of wild-type p53 expression. V type lung cancer cells undergo, as described in previous reports, cell senescence and stop proliferating. In contrast, D type lung cancer cells rapidly die, not by undergoing apoptosis but through a cyclophilin dependent (CsA inhibitable) necrotic death. The authors present RNAseq data and data from screens to try to identify the underlying mechanisms that determine whether lung cancer cells behave as type V or type D.

Many of the results shown come from high quality experiments and the findings are interesting. However, additional experiments need to be conducted to provide stronger support for some of the conclusions brought forward by the authors.

Figure 2: Analysis at 3 and 14 days after p53 re-expression may be too late to see apoptosis in type D lung cancer cells. I suggest to examine after 1 day of restoration of wild-type p53 expression, using staining for cleaved (i.e activated) caspase-3, which is more specific for detecting apoptosis than TUNEL staining, which also labels cells dying by processes other than apoptosis.

The authors should examine whether the appearance of areas of necrosis in type D lung cancer cells after restoration of wild-type p53 expression is mediated by immune system activation, responding to cellular senescence induced by p53 reactivation. This could be done by transplanting type D tumours into immune-deficient mice and mice with a competent immune system and then restore p53 in these malignant cells. If areas of necrosis will only appear after restoration of wild-type p53 expression in recipient mice with a normal immune system, then the immune system is critical for this.

Figure 4: To block apoptosis, both BAX and BAK need to be removed. The authors should generate BAX/BAK double knockout type D lung cancer cell lines and examine their response to restoration of wild-type p53 expression. If the cells still die, this will prove beyond doubt that apoptosis is not critical for their killing.

The authors should also examine whether the death of D type lung cancer cells after restoration of wild-type p53 expression is due to autophagy associated death. This can be tested by using Crispr to delete a gene essential for autophagy, such as Atg7.

Clonogenic cell survival assays should be performed to test whether treatment with CsA allows D type lung cancer cells to survive AND continue to proliferate after restoration of wild-type p53 expression.

Interaction of p53 with cyclophilin A must be demonstrated by co-IP experiments with endogenously expressed proteins and not only with over-expressed proteins.

The function of the genes that are impacted by CsA treatment in the D type lung cancer cells after restoration of wild-type p53 expression must be examined by induced over-expression of these genes or Crispr mediated deletion of these genes. Are some of these genes essential for cell killing after restoration of wild-type p53 expression in D type lung cancer cells? What happens when these genes are over-expressed or deleted in V type cells after restoration of wild-type p53 expression.

Figure 1G: the numbers of mice of each genotype examined should be shown in the figure itself for clarity.

Figure 2B: the numbers of mice examined should be increased.

All figures showing Western blots or IP Western blots need to indicate molecular weight markers on the side of the blots.

Figure 3B and Supplementary Figure S3: I am surprised at how high the levels of wild-type p53 protein are in the cells after restoration of a functional wild-type p53 allele. The levels of wild-type p53 are usually very low in unstressed cells. What is the explanation for the high levels of wild-type p53 protein? Are these cells stressed?

RESPONSE TO REVIEWERS' COMMENTS

ALL REFEREES: We thank the reviewers for their interest in our study and their suggestions on how to improve the quality of our manuscript. Most of the reviewer suggestions focused on: providing controls to support relevant experimental conclusions or knockdown efficiencies, additional insight into endogenous cyclophilin-p53 interactions, measuring cyclophilin protein levels in our models, determining the role of the immune system and other forms of cell death, and the translational implications of cyclophilin-dependent death in human SCLC. The reviewers also identified areas where data presentation could be improved to aid in the interpretation of our study. In total we have added more than 60 new figure panels to the manuscript to address the reviewers concerns. Additional data that addresses reviewer comments but does not fit neatly into the manuscript is also appended below in the relevant section. We believe these additions enhance the rigor, quality, and clarity of our paper. We thank the reviewers for the suggestions.

The specific areas of concern indicated by each referee are listed below followed by a detailed response for each:

REFEREE 1:

MAJOR POINTS

Q1.1: *“While the cyclosporin A rescue of cell death in vitro upon p53 restoration provides a reasonably convincing starting point for a cyclophilin dependent death mechanism, the genetic rescue by sgRNA deletion of cyclophilin A and E lacks several important controls that will help explain variable and/or phenotypically weak results. In the initial sgRNA experiment and validation in different cell lines (Fig4E-G) the effect of targeting Ppia provides either dubious rescue (4F) or a highly variable one (4G). Is this because of variable editing efficiency or cell line specific/contextual roles for the different cyclophilins? The authors should show western blots and/or sequencing evidence demonstrating the level of knockdown or editing efficiency corresponding to the degree to which competitive fitness or viability is rescued.”*

A1.1: Thank you for highlighting this. In the revised manuscript we have added additional sgRNAs targeting *Ppia* and provided western blots to demonstrate the relative effect for each on CypA protein levels (**Figures 4C, S11A, and S11B**). Variability that the reviewer points out comes from multiple sources. As suggested, there is certainly variation in targeting efficiency between guides. There is also variation in effect size between biological replicates (*i.e.* independent SCLC cell lines). Finally, the assay of mCherry enrichment has intrinsic variability due to the dependency on lentiviral transduction efficiency between technical and biological replicates. Despite the many sources of variability, in every replicate (biological or technical) we see enrichment for sgRNAs targeting *Ppia* and *Ppie*. This robustness demonstrates that these two genes are important for Type D SCLC death.

Q1.2: *Given the ability of one of the Ppie sgRNAs to rescue to a similar extent as sg-p53 in 4G, the authors should generate a series of single and double Ppia and Ppie knockout “D” lines with accompanying analysis of both knockout efficiency and viability*

A1.2: This is a great suggestion but happens to be trickier than expected. It has been impossible for us to isolate cells that lack CypA or CypE entirely. Significant reduction of CypA seems to be tolerated and we are able to isolate stable lines that have ~90% reduction in CypA protein levels. In contrast, loss of CypE seems not to be tolerated at all (**Rebuttal Figure A**). Although we see significant reduction of CypE in the first few days after transduction of sgRNAs targeting *Ppie*, the cells that emerge after serial passage have normal CypE expression levels (**Figure S18A**). This requirement of CypE expression has limited some analyses. However, we were able to use the CypA *reduced* cells to subsequently target CypE using the transient

mCherry enrichment assay. In this scenario, reduction of CypA had a partial effect blunting p53-mediated Type D cell death. The additional transient targeting of CypE had an additive effect that phenocopied that of targeting p53. Thus, we interpret the data to mean that both CypA and CypE contribute to p53-mediated TypeD cell death in separate ways and the effects of targeting both may abrogate p53's ability to engage this cell death program (**Figures 4F and S11C**). Mechanistically, in newly added data we show that the Type D cell death gene expression module (Module 5) is at least partially dependent on the expression of CypA (**Figures 6K and 6L, Figure S18**). The requirement for maintained CypE expression did not allow us to examine the combined effects of targeting both genes using the bulk RNA-sequencing approach.

Rebuttal Figure A (Reviewer 1). Cyclophilin E expression is lost over time in selected cells. Representative histograms of selected BGal (LentiCRISPRv2-mCherry), and CypA (LentiCRISPRv2-Puro) /CypE (LentiCRISPRv2-mCherry) double knockout cells. mCherry expression levels were overlaid over Day 0 after 7 days in culture. Experiment was conducted at least 2 weeks after selection.

Q1.3: “Another important control that I could not find is the effect of cyclophilin knockout in “V” type cells. Presumably their knockout would not alter the trajectory upon p53 restoration in these cells”

A1.3: The reviewers presumption is correct! Thus, we have included data in (**Figures S6A and 4E**) that address the effect of pharmacological inhibition or genetic knockout of cyclophilins in Type V cells. As expected, cyclophilin inhibition does not alter the trajectory upon p53 restoration in Type V SCLC.

Q1.4: “Furthermore, although baseline cyclophilin gene expression (Fig S8) does not explain the “V” versus “D” phenotype, it is not clear if different cyclophilin protein levels and distinct patterns of cyclophilin binding can explain the distinct p53-dependent biology in the 2 classes of cells. Unless I missed it, the binding experiments were all performed in an overexpression setting in 293T cells rather than with endogenous proteins in the primary cell lines. The protein levels of cyclophilins in the different classes of cell lines as well as the ability of endogenous p53 and cyclophilins to bind should be tested.”

A1.4: We have incorporated new data into our study that addresses these important questions. In fact we believe this is the most important and exciting addition to the paper. First we provide western blot analyses in 3 Type D and 3 Type V cells lines for basal CypA and CypE expression (**Figure S14A**). Additionally, we provide additional analysis of changes in CypA and CypE expression after p53 restoration and CsA treatment in representative Type D and Type V cells (**Figure S14B**). Protein levels do not vary appreciably between the cell types or from each treatment. Importantly, in a new collaboration with Dr. Maureen Murphy, we conducted proximity ligation assays to determine the extent to which endogenous cyclophilins interact with endogenous p53 either before or after p53 restoration and determined how these interactions are affected by CsA. Consistent with the over expression experiments, in 293T cells, we detect p53 interactions with CypA but not CypE. Moreover, this interaction is disrupted by CsA. Most interestingly, the CypA:p53 interaction occurs ~10 fold more frequently in the Type D cell lines compared to the Type V cell lines (**Figures 4I and 4J**). We do not know why this distinction exists, but its specificity for the Type D cell type is most intriguing and fits the model that p53 promotes a unique gene expression program in Type D cells that is dependent on the CypA:p53

interaction and that this gene expression program ultimately leads to the novel form of cell death we describe.

Q1.5 *“The authors propose that p53-cyclophilin binding directs the “D” versus “V” phenotype due to the ability of cyclosporin A to blunt “D” specific genes upon p53 restoration. 2 important experiments are missing to properly interpret this. First, p53 accumulation must be measured in the setting of cyclosporin A treatment and in genetic experiments with specific cyclophilin knockdown. Can cyclophilin binding alter p53 stability? Both p53 hallmark genes and the cluster 5 genes are reduced in restored cells with cyclosporin A treatment, along with a modest reduction in ChIP signal. This would be consistent with reduced p53 levels in cells.”*

A1.5: We have included additional data in **Figures S14B and S14C** that address the effect of cyclophilin inhibition or knockdown on p53 stability. As shown, we did not observe a meaningful decrease in p53 protein stability across SCLC subtypes after p53 restoration, suggesting that differential regulation of p53 protein is likely not the mechanism by which cyclophilins are decreasing p53 gene expression. Additionally, it should be highlighted that a very modest reduction in p53 ChIP signal was only observed in Type V cells treated with CsA where no decrease in p53-target gene expression was detected (**Figure 6E**). In contrast, p53 ChIP signal was actually slightly higher in Type D cells treated with CsA where we observe the reduction in p53 target gene expression (**Figures 6D, 6I, and 6J**). These data suggest that cyclophilin-mediated regulation of p53 protein levels are not likely to be the mechanism by which Type D cells survive after CsA treatment.

Q1.6: *“Also, as mentioned in point 1, the same sgRNA system with validated knockout/editing efficiency should be used to validate cyclophilin dependency of these gene expression changes. RNA SEQ is not necessary, but a set of presumably cyclophilin dependent genes should be assayed up p53 restoration in sgPpia, sgPpie, and double sgPpia/Ppie cells. Again, this would demonstrate the cyclophilin specificity of this phenotype.”*

A1.6: As stated in the answer to Point 1 above, we did perform RNA-seq on CypA reduced cells following p53 restoration. As shown in **Figures 6K, 6L, and S18**, the ~90% reduction of CypA leads to a significant decrease in the expression of p53-target genes and Module 5 genes. Due to the biological constraints of the strict requirement for CypE expression, we were unable to test its effect on transcriptional output. However, that CypA depletion does at least partially restrict p53-mediated transcriptional output strongly supports the model that cyclophilins are needed to support p53-mediated transcription in these Type D cells.

Q1.7: *“In vivo rescue of regressions by cyclosporin A is a compelling result, but could be due to non-specific effects of the drug on tumor cells and the microenvironment. The authors should perform the allograft p53 restoration experiments shown in Figure 3K with a set of “D” cell lines expressing sgRNAs targeting Ppia, Ppie, and both sgPpia/Ppie. This will address if the cell death phenotype in vivo is indeed cyclophilin specific and cell intrinsic.”*

A1.7: We thank the reviewer for highlighting a potential role of the microenvironment on Type D cell death and the effects of CsA. All of the data refute prominent roles of the microenvironment in the regression of Type D SCLC or an off-target effect of CsA on tumor regressions in vivo. We demonstrate that all Type D SCLC cell lines regress upon p53 restoration in severely immunocompromised mice (**Figure S8**). We show that changes in immune cell frequency is limited after p53 restoration in the lungs and in lung tumors in an autochthonous model where the immune system is fully intact (**Figure S8A and S8B**). While there could be some effect of CsA treatment on immune cells in the tumor microenvironment, the negligible role of the immune system for tumor regression strongly suggests this effect is not a major one. Taken together with the extensive in vitro data that exclude a role of immune cells in this cancer cell

autonomous mechanism of p53-mediated Type D cell death, as well as the new RNA-seq data that demonstrates a CypA dependency for p53-mediated gene expression, we felt these additional transplant experiments were not likely to yield any new insight.

MINOR POINTS

Q1.8: “The author’s use of a spectrum of well-established inhibitors of cell death is a convincing approach to identify specific mechanisms by which “D” cells die. They state that while p53 restoration results in cyclosporin A sensitive necrosis, it is not characterized by notable post-translation activity of p53 at the mitochondria (e.g. Vaseva et al., 2012). However, there does appear to be more p53 in the mitochondrial fraction of the “D” versus “V” cells shown in S5I. Is this observed consistently across additional “D” and “V” cell lines? To better rule out the mPTP route the authors should gauge another functional characteristic of that path to cell death, such as mitochondrial swelling, or perform an orthogonal, more sensitive, test for p53 localization (e.g. co-staining of p53 and mitochondria).

A1.8: We acknowledge that the potential mitochondrial functions of p53 needed to be better addressed in our study and we thank the reviewer for the suggestion. First, we used proximity ligation assays to detect p53 co-localization with the mitochondrial protein Tomm20. p53 interacts with Tomm20 to a small extent in both Type V and Type D cells and these interactions are not affected by the addition of CsA (**Figures S13B-S13E**). We also determined CypD and p53 protein interactions. As shown in **Figure S12**, p53 can interact with CypD in both Type D and Type V cells in a manner that can be disrupted by CsA. However, similarly to Tomm20, these interactions are quite limited in Type V and Type D cells. We have also provided additional evidence that CRISPR knockout of CypD (*Ppif*) does not block p53-mediated death and validated the efficacy of the sgRNAs (**Figures 12E and 12F**).

Q1.9: IHC for p53 and representative direct targets should be performed in tissues after p53 restoration. While tamoxifen treatment clearly alters the growth of tumors, the extent of p53 accumulation is associated with restoration associated phenotypes in vivo (notably in mutant Kras driven lung cancers). Is heterogeneity in vivo also due differences in p53 accumulation or activity as measured by a target like p21?

A1.9: This is unfortunately technically challenging. Detecting wildtype p53 or p21 in mouse IHC has not been possible. Although in the past, I have been able to see faint expression of p53 or p21 in lung adenocarcinoma models after p53 restoration (Feldser et al. Nature 2010), here we have been unable to see this. In both cases, the exact antibody reagents are no longer available. What we can see is that loss of GFP expression from the gene trap is widespread at early time points after p53 restoration (**Figure 1F**). However, there are most definitely cells in these SCLC that do not effectively excise the gene trap after tamoxifen treatment as intended. We know this because all mice that succumb to SCLC after p53 restoration (which occurs many weeks later) have bright GFP-positive tumors indicating an outgrowth of SCLC cells that have an active gene trap and therefore no p53 expression (**Figure S1**). It should be noted however, that the heterogeneity is not between tumors per se but within each tumor. That is to say, all tumors lose GFP expression to the same extent early, and all tumors that grow out in the end are GFP positive just like tumors where we never restored p53 at all.

Q1.10: In Figure 5, the authors should label the matched control versus restored pairs of “D” and “V” cells. It is unclear if p53 restoration in the “V” cells has a coherent effect in PC2 as it does in the “D” cells.

A1.10: We thank you for point out that our PCA analysis was difficult to interpret. We have labeled the matched control:restored pairs by connecting the samples with dotted lines.

Q1.11: There are several typos and fragmented sentences throughout.

A1.11: We have proofread our manuscript to eliminate any typos and fragmented sentences.

REFEREE 2

Q2.1: The major weaknesses include lack of clear relevance to SCLC therapy, as they predict that CsA would block these therapies, no strategy to enhance efficacy of restoration is proposed, and no effective p53 restoration therapeutics are on the horizon. Were there any clues from the analysis of Type V cells that suggests specific therapeutic vulnerabilities?

A2.1: We thank the reviewer for identifying that our paper does not immediately shape therapeutic strategies for SCLC. We believe that is a lofty goal and one that we never intended to achieve. Our goal was to understand the biology of p53 in the context of SCLC. In so doing we have discovered a new form of cell death that can be regulated by p53 and we have made significant progress toward understanding its mechanics in this manuscript. I do believe that a complete understanding of this cell death program has the potential to identify therapeutic strategies for SCLC or other cancer types but this is well beyond the scope of this paper. The reviewers comment did arouse our curiosity for whether we could coax Type V cells into Type D cells with clinically meaningful approaches. We therefore tested whether stabilizing p53 with Nutlin-3a could promote p53-mediated death in Type V cells that otherwise would undergo senescence. As shown in **Figure S6**, Nutlin-3a treatment significantly enhances the ability of p53 to induce cell death in both Type D and Type V cells. Most importantly, CsA blocks p53-mediated death even after Nutlin-3a treatment in both Type D and Type V cells. This insight has identified a specific vulnerability in Type V cells to p53 stabilizing drugs and shows that if/when a p53 mutant-reactivating drug becomes available that combination with Nutlin3a may enhance its tumor cell killing effects.

Q2.2: *“Also notably, p53 restoration in human SCLC-A and SCLC-N cell lines was not shown to corroborate these findings”*

A2.2: We addressed this by re-expressing wild-type p53 in human SCLC cells: ASCL1+ (H69, H2081, H2195) and NEUROD1+ (HCC4004, HCC2433) using adenoviral vectors. Re-introduction of wild-type p53 in human SCLC cell lines recapitulated the heterogeneity that we describe in our study. A subset of cell lines underwent p53-mediated cell death that was exacerbated by Nutlin-3a treatment. In one cell line that was most highly enriched for the Type D gene expression signature, the death-inducing effect of p53 re-introduction was blocked by CsA. On the other hand, the NEUROD1+ subset of human cell lines did not undergo cell death after wild-type p53 introduction and were also not impacted by the addition of Nutlin3a. While neither of the two NEUROD1+ cell lines died in our experiment, one line did express significant levels of mutant p53 (the other lines did not) that likely impacted the ability of wildtype p53 to perform (**Figures 5D-5P**). Although future studies will be needed to further understand the role of p53 in distinct SCLC subtypes in human and mouse model systems, we believe that the experiments now included in our study support a context-dependent function of p53 in SCLC.

Q2.3 *“Figure 2D: Ki67+ of ~20% seems very low for human SCLC”*

A2.3: Clearly differences exist between mouse and human SCLC. Perhaps the extent of Ki67 expression is linked to the advanced state of disease when detected in a patient? Our ability to anticipate disease onset and monitor its growth earlier in the mouse may contrast that which is observed in a clinical setting. Regardless, these mouse SCLC are aggressive and fast growing.

REFEREE 3

Q3.1: *“Figure 2: Analysis at 3 and 14 days after p53 re-expression may be too late to see apoptosis in type D lung cancer cells. I suggest to examine after 1 day of restoration of wild-type p53 expression, using staining for cleaved (i.e activated) caspase-3, which is more specific for detecting apoptosis than TUNEL staining, which also labels cells dying by processes other than apoptosis.”*

A3.1: We expanded our *in vitro* cell death analysis in Type D cells to now include 24hr protein analysis of cleaved Caspase 3 expression. Consistent with our data from later timepoints, we do not see cleavage of Caspase 3, as shown in **Figure S4E**.

Q3.2: *“The authors should examine whether the appearance of areas of necrosis in type D lung cancer cells after restoration of wild-type p53 expression is mediated by immune system activation, responding to cellular senescence induced by p53 reactivation. This could be done by transplanting type D tumours into immune-deficient mice and mice with a competent immune system and then restore p53 in these malignant cells. If areas of necrosis will only appear after restoration of wild-type p53 expression in recipient mice with a normal immune system, then the immune system is critical for this.”*

A3.2: The role of the immune system in Type D cell death is an important factor to account for given the previously described role of the innate immune system in clearing tumor cells after p53 reactivation that we and others have described in other mouse models (Ventura et. Al. Nature 2007, Xue et. al. Nature 2007, Feldser et al. Nature 2010, Stokes et. al. Oncogenesis 2019). However, several lines of investigation suggest that the immune system does not play a prominent role in this context. First, we showed that 4 independent Type D cell lines regress after p53 restoration when transplanted in immune incompetent mice (**Figures S8C-S8G**). Moreover, as p53-mediated necrosis occurs *in vitro*, we can safely conclude that Type D cell death is a cancer cell autonomous process that proceeds in the absence of immune cells. To better highlight these data, we have incorporated changes in the text to clarify that our transplant experiments were conducted in immunodeficient animals.

Q3.3: *“Figure 4: To block apoptosis, both BAX and BAK need to be removed. The authors should generate BAX/BAK double knockout type D lung cancer cell lines and examine their response to restoration of wild-type p53 expression. If the cells still die, this will prove beyond doubt that apoptosis is not critical for their killing.”*

A3.3: We agree that it is imperative that we rule out apoptosis as the mechanism for p53-mediated death. While Bax and Bak double knockout is an excellent approach to address this, we decided to employ the orthogonal approach of overexpressing the anti-apoptotic protein, Bcl-2, in Type D cells. This strategy allowed us to streamline the revision process and avoid having to apply multiple selective pressures on the cells to generate Bax:Bak double knockout cells. We show in **Figures S4F-S4I** that Bcl-2 over-expression is unable to prevent p53-mediated death in Type D cells after p53 reactivation. We believe that, in combination with the lack of Caspase 3 cleavage, lack of TUNEL staining, and no effect of ZVAD-FMK, we have convincingly shown that p53 is not inducing apoptosis in Type D SCLC cells.

Q3.4: *“The authors should also examine whether the death of D type lung cancer cells after restoration of wild-type p53 expression is due to autophagy associated death. This can be tested by using Crispr to delete a gene essential for autophagy, such as Atg7.”*

A3.4: To further rule out other mechanisms of cell death, we knocked out Atg5 in multiple Type D cell lines. This choice was again made out of convenience because our laboratory neighbors had mouse Atg5-targeting lentiviral plasmids that we could immediately use in our mCherry enrichment assay. Consistent with autophagy not playing a causal role during p53-mediated death in Type D cells, Atg5 knockout mCherry⁺ cells were not enriched after p53 restoration (**Figure S10**). Taken together, these data suggest that autophagic cell death is not induced by p53 to cull Type D cells.

Q3.5: *“Clonogenic cell survival assays should be performed to test whether treatment with CsA allows D type lung cancer cells to survive AND continue to proliferate after restoration of wild-type p53 expression.”*

A3.5: While clonogenic cell survival assays are an excellent qualitative approach to assess cell proliferation, we decided to thoroughly address this reviewer comment by conducting BrdU incorporation assays to quantify cell cycle progression and measure the onset of senescence via SA- β Gal staining. CsA-treated Type D cells are arrested in the cell cycle even though they do not undergo p53-mediated death. Interestingly, these Type D cells also did not have significant SA- β Gal staining perhaps indicating that they are not fully senescent either. These data suggest that cyclophilin inhibition does not convert Type D cells into senescing Type V cells, but p53 can induce a non-proliferative cell state when Type D cell death is blocked. These new data are located in **Figure S5**.

Q3.6: *“Interaction of p53 with cyclophilin A must be demonstrated by co-IP experiments with endogenously expressed proteins and not only with over-expressed proteins.”*

A3.6: (copied from A1.4 above) In a new collaboration with Dr. Maureen Murphy, we conducted proximity ligation assays to determine the extent to which endogenous cyclophilins interact with endogenous p53 either before or after p53 restoration and determined how these interactions are affected by CsA. Consistent with the over expression experiments, in 293T cells, we detect p53 interactions with CypA but not CypE. Moreover, this interaction is disrupted by CsA. Most interestingly, the CypA:p53 interaction occurs ~10 fold more frequently in the Type D cell lines compared to the Type V cell lines (**Figures 4I and 4J**). We do not know why this distinction exists, but its specificity for the Type D cell type is most intriguing and fits the model that p53 promotes a unique gene expression program in Type D cells that is dependent on the CypA:p53 interaction and this gene expression program ultimately leads to a novel form of cell death.

Q3.7: *“The function of the genes that are impacted by CsA treatment in the D type lung cancer cells after restoration of wild-type p53 expression must be examined by induced over-expression of these genes or Crispr mediated deletion of these genes. Are some of these genes essential for cell killing after restoration of wild-type p53 expression in D type lung cancer cells? What happens when these genes are over-expressed or deleted in V type cells after restoration of wild-type p53 expression”*

A3.7: We share the reviewers enthusiasm for better understanding the genetic determinants of Type D cell death. To be certain, we are following up this study with the exact approaches the reviewer outlines. However, this will form the basis of a new project and is too massive of an undertaking to reasonably include here.

Q3.8: *“Figure 1G: the numbers of mice of each genotype examined should be shown in the figure itself for clarity”*

A3.8: We have corrected the figure to include the number of mice.

Q3.9: “Figure 2B: the numbers of mice examined should be increased”

A3.9: We have already included the maximum amount of animals available to the data in **Figure 2B**. However, the data in **Figure 3M** expands on our analysis and demonstrates that the result is reproducible across distinct experiments. We have combined the data from both experiments here for the reviewer’s convenience (**Rebuttal Figure B**).

Q3.10: “All figures showing Western blots or IP Western blots need to indicate molecular weight markers on the side of the blots.”

Rebuttal Figure B (Reviewer 3). Comprehensive analysis of uCT data. Data from Figures 2B and 3M were combined to increase the number of animals in the analysis.

A3.10: We have added the appropriate molecular weight markers to all of our western blots.

Q3.11: “Figure 3B and Supplementary Figure S3: I am surprised at how high the levels of wild-type p53 protein are in the cells after restoration of a functional wild-type p53 allele. The levels of wild-type p53 are usually very low in unstressed cells. What is the explanation for the high levels of wild-type p53 protein? Are these cells stressed?”

A3.11: We established cell lines from advanced SCLC tumors. It is expected that cancer cells that are highly proliferative will be under high levels of cellular stress (e.g. replicative, metabolic, culture etc.) to support unregulated growth. A common feature of p53 deficient cancer cells is expression of the p19/ARF tumor suppressor—a negative regulator of Mdm2. We show in **Figure S2E** that SCLC cells express p19/ARF which would promote wildtype p53 stabilization upon restoration.

REVIEWER COMMENTS

Reviewer #1 (Remarks to the Author):

The revisions have addressed most of the concerns from the original submission. Of note, the PLA data demonstrating increased interaction of p53 and Ppia in D versus V cells and p53 re-expression in human SCLC lines resulting in lineage associated cyclosporin sensitive cell death significantly improves the manuscript.

There are 2 remaining concerns regarding the findings:

1. The authors performed western blots to demonstrate that reduced levels of cell death upon p53 restoration correlates with the levels of cyclophilin A and E in sgRNA edited D cells (Supplemental Figure 8A-C). Is this the same primary cell line used in Figure 4? The viability rescue in Figure 4 does not correspond with the protein levels associated with the guides in Supplemental Figure 8 (e.g. sgPpia3.2 is associated with lowest protein level, highest rescue (Sup Figure 11), versus lowest degree of rescue in Figure 3C, with similar disconnects with sgPpie7.1 and 10.1 which in the supplement show similar degree of efficiency of knockdown, but dramatically different rescue in Figure 3D). Presumably this is due to different levels of editing in different primary cell lines. While it would be most convincing if western blots matching the competition assays were matched, the authors should make it clear in the figures and legends which primary cell line is being used for each experiment. This is critical for example in Supplemental Figure 8B where repeat blots for Ppie in sgBgal, sgPpi37.1 and 10.1 labeled lanes are shown with very different protein levels in each series. Are these different primary cell lines? Is the effect of each sgRNA consistent in each primary cell line it is used in repeat experiments?

2. While the work now more persuasively demonstrates the role that cyclophilin A and E play in Type D cell death upon p53 restoration in vitro, the evidence for their role in vivo is still complicated by potential off-target effects of cyclosporin. The aim of comment 1.7 was to confirm that cyclophilin A/E are cell intrinsic triggers of p53 induced cell death in vivo. While the authors have now clarified that nude mice were used as hosts in the allograft experiments shown in Supplemental Figure 8, this data suggests that an entirely intact adaptive immune system is not necessary for regression of tumors consisting of type D cells. However, nude mice retain macrophages that can contribute to regression of p53 reactivated tumors (Xue et al., 2007). As macrophages are the notable cell type that infiltrates into Type D like tumors that develop necrosis and fibrosis, it is possible that they are playing a role in this allograft model. Unless I missed it there is no comparison of allograft type D cells in different immune backgrounds, e.g. Nude versus immune intact, or nude versus more compromised such as NSG, making it difficult to conclusively rule out cell extrinsic effects.

An experiment testing if stable KO/knockdown on Ppia or Ppie in the allograft setting prevents or blunts regressions would alleviate concerns regarding cyclosporin artifacts and better address potentially relevant effects of the microenvironments. Given the consistent in vitro results in these genetic experiments however, I would suggest the authors temper in vivo implications of the cyclophilins to "cyclosporin sensitive" and more carefully describe the interpretations of the allograft experiments, recognizing that possible immune effects were not exhaustively dissected.

One minor point that should be addressed (as mentioned in point 1 above), the authors should carefully review figure and legend labeling. Not only should the identity of primary cell lines be updated but care should be taken for axis and graph labeling. For example, the tornado plots for ChIP seq are labeled as "genes" when this should be "peaks", unless all of these peaks are in gene bodies.

Reviewer #3 (Remarks to the Author):

While the authors have performed some of the experiments that I have requested when reviewing the original version of this paper, they did not perform a lot of important work that I also asked

for. In my opinion, in the absence of data from these requested experiments, this paper is not of the quality that I expect for a paper published in Nature Communications.

Point 3.5: Colony formation assays should have been performed.

Point 3.6: very important! co-IP of endogenous p53 with endogenous cyclophilin must be shown to demonstrate the interaction of these two proteins under physiological conditions rather than using ectopic over-expression (as the authors did), which is widely known to cause artefacts.

Point 3.7: very important! at least two hits should have been examined by either over-expression or deletion using CRISPR to reveal the mechanism of tumour growth inhibition and tumour cell killing.

RESPONSE TO REVIEWERS' COMMENTS

ALL REFEREES: We thank the reviewers for their continued interest in our study and their suggestions on how to further improve the quality of our manuscript. We have addressed the remaining concerns below point by point.

REFEREE 1:

Q1.1: The authors performed western blots to demonstrate that reduced levels of cell death upon p53 restoration correlates with the levels of cyclophilin A and E in sgRNA edited D cells (Supplemental Figure 8A-C). Is this the same primary cell line used in Figure 4? The viability rescue in Figure 4 does not correspond with the protein levels associated with the guides in Supplemental Figure 8 (e.g. sgPpia3.2 is associated with lowest protein level, highest rescue (Sup Figure 11), versus lowest degree of rescue in Figure 4[sic]C, with similar disconnects with sgPpie7.1 and 10.1 which in the supplement show similar degree of efficiency of knockdown, but dramatically different rescue in Figure 3D). Presumably this is due to different levels of editing in different primary cell lines. While it would be most convincing if western blots matching the competition assays were matched, the authors should make it clear in the figures and legends which primary cell line is being used for each experiment. This is critical for example in Supplemental Figure 8B where repeat blots for Ppie in sgBgal, sgPpi37.1 and 10.1 labeled lanes are shown with very different protein levels in each series. Are these different primary cell lines? Is the effect of each sgRNA consistent in each primary cell line it is used in repeat experiments?

A1.1 Thank you for highlighting this confusion. We believe the reviewer is referring to Supplementary Figure 11 given that Supplementary Figures 8A-8C do not discuss CypA and CypE sgRNA edited cells, but rather have data regarding the role of the immune system in autochthonous and allograft SCLC tumors. The cell line used in **Supplementary Figures 11A and 11C (4711-1)** is not the same as the one used in Figure 4 (**4B: 4716-11 or 4C,4D, and 4F: 4711-18**). Unfortunately, we only have matched western blot data for some of the competition experiments. We completely agree with the reviewer that changing the labels to state which cell line was used will make the data easier to interpret. We incorporated these changes throughout the manuscript as suggested. Adding these labels clarifies to the reader that the data depicted are from distinct cell lines, not different experiments.

Regarding correlation of knockdown and effect size of survival rescue: The short answer is that the data in Figure 4 and those in SFig11 are not matched and are from independent experiments. Despite this, the data indicate that the degree of Ppie knockout is generally correlated with the effect size of survival rescue across these independent experiments. For example sgRNA Ppie#7.1 has the most protective effect and best knockdown (Figure 4D and SFig 11B). However, as the reviewer pointed out, the data suggest that the degree of Ppia knockdown does not correlate with the degree of the protective effect of each guide across these independent experiments. However, because the data in Figure 4C and SFig 11A are not matched, caution should be taken to assume that the degree of knock down is the same between experiments given that we did not assess protein levels at all in Fig4. Importantly, we note that the relative enrichment for independent sgRNAs targeting *Ppia* across each Type D cell line and across each experimental replicate is reproducible. We provide an additional experiment here that was not included in the paper to show that reproducibility. Again, we see that sgPpia4.1 is the most protective guide against p53-mediated TypeD cell death (1.9 fold enrichment) and sgPpia3.2 is slightly less protective against p53-mediated TypeD cell death (1.5 fold enrichment). These values are highly similar to those shown in Fig4C where sgPpia4.1 was enriched 1.9 fold and sgPpia3.2 was enriched 1.3 fold. Overall, the data certainly indicate that cyclophilin A and E are key effectors of p53-mediated death in SCLC and the extremely slight differences related to cell protective effects between sgRNAs is of minor concern.

FigureA1.1. Enrichment assay for sgPpia expressing vectors. Cell line is 4711-1.

Q1.2: While the work now more persuasively demonstrates the role that cyclophilin A and E play in Type D cell death upon p53 restoration in vitro, the evidence for their role in vivo is still complicated by potential off-target effects of cyclosporin. The aim of comment 1.7 was to confirm that cyclophilin A/E are cell intrinsic triggers of p53 induced cell death in vivo. While the authors have now clarified that nude mice were used as hosts in the allograft experiments shown in Supplemental Figure 8, this data suggests that an entirely intact adaptive immune system is not necessary for regression of tumors consisting of type D cells. However, nude mice retain macrophages that can contribute to regression of p53 reactivated tumors (Xue et al., 2007). As macrophages

are the notable cell type that infiltrates into Type D like tumors that develop necrosis and fibrosis, it is possible that they are playing a role in this allograft model. Unless I missed it there is no comparison of allograft type D cells in different immune backgrounds, e.g. Nude versus immune intact, or nude versus more compromised such as NSG, making it difficult to conclusively rule out cell extrinsic effects.

An experiment testing if stable KO/knockdown on Ppia or Ppie in the allograft setting prevents or blunts regressions would alleviate concerns regarding cyclosporin artifacts and better address potentially relevant effects of the microenvironments. Given the consistent in vitro results in these genetic experiments however, I would suggest the authors temper in vivo implications of the cyclophilins to “cyclosporin sensitive” and more carefully describe the interpretations of the allograft experiments, recognizing that possible immune effects were not exhaustively dissected.

A1.2: To be brief, we have tempered the language throughout the manuscript as suggested by the reviewer. We agree that this more exact language better reflects the findings.

As stated in our previous response, it has not been possible for us to generate Type D cells that have sustained loss of CypA and CypE expression because long term KO of these proteins seems to decrease cell fitness (Supplementary Figure 18). As such, this poses a significant roadblock for allograft experiments that use CRISPR-modified Type D cells, since they regain (or rather select for cells that have not lost) expression of the targeted cyclophilin during the experimental timeframe. While we are interested in studying the effect of CypA and CypE loss in an autochthonous setting, generating an *in vivo* SCLC model to study this will be an extremely arduous task. We agree with the reviewer that macrophage infiltration into necrotic and/or fibrotic regions complicates the interpretation of the *in vivo* results since it is difficult to tease apart whether or not the macrophages are directly killing tumor cells, or simply clearing dead cell debris. However, we feel that this is a relatively minor concern given that the combination of all the data strongly argues against any requirement of immune cells, or any other cell type, for promoting the death of Type D cells. Specifically, we show:

1. Type D cell death occurs in cell culture in the absence of any other cell type,
2. Genetic perturbation of cyclophilins A and E or pharmacological inhibition of cyclophilins blocks Type D cell death in the absence of any immune cells *in vitro*,
3. Though an immune cell-associated mechanism of action for CsA through T cells exists, we show that T cells are dispensable for death of Type D cells upon p53 restoration *in vivo*,
4. We show that cyclophilin inhibitors that do not impact NFAT signaling (the mechanism of CsA action in T cells) are equally potent to limit Type D cell death, again in the absence of any immune cells *in vitro*, and
5. The work identified by the reviewer (Xue et al. Nature 2007) implicating innate immune cells in the regression p53-reactivated tumors was associated with tumors that are senescent, a phenotype that is not occurring in Type D SCLC.

Q1.3: One minor point that should be addressed (as mentioned in point 1 above), the authors should carefully review figure and legend labeling. Not only should the identity of primary cell lines be updated but care should be taken for axis and graph labeling. For example, the tornado plots for ChIP seq are labeled as “genes” when this should be “peaks”, unless all of these peaks are in gene bodies.

A1.3: We revised and modified the figures and legends, as suggested by the reviewer, to enhance the clarity of our study.

REFEREE 3:

Q3.1: Colony formation assays should have been performed.

A3.1: Reviewer 3 was interested in determining the extent to which CsA-treated Type D cells continue to proliferate after surviving p53-mediated death. While we did not conduct colony formation assays, we explained that we extensively addressed their concern by conducting BrdU incorporation assays to quantify cell cycle progression and measure the onset of senescence via SA-βGal staining (Supplementary Figure 5). As stated in the original rebuttal letter, CsA-treated Type D cells are arrested in the cell cycle even though they do not undergo p53-mediated death. Interestingly, these Type D cells also did not have significant SA-βGal staining perhaps indicating that they are not able to easily engage the senescence program either. We believe that these orthogonal approaches address the reviewer’s concerns in an even more stringent manner and shed light on cell cycle and senescence mechanisms.

Q3.2: very important! co-IP of endogenous p53 with endogenous cyclophilin must be shown to demonstrate the interaction of these two proteins under physiological conditions rather than using ectopic over-expression (as the authors did), which is widely known to cause artifacts.

A3.2: We collaborated with Dr. Maureen Murphy's laboratory to assess endogenous interactions between p53 and cyclophilins using Proximity Ligation Assays (PLA). No target protein was over expressed in these assays, thus eliminating the impact of potential artifacts in the conclusions drawn from our study. As shown in Figures 4I and 4J, p53 endogenously interacts with cyclophilin A but not cyclophilin E, consistent with the findings from the exogenous protein expression experiments. Furthermore, PLA assays give additional information about the cellular location of the interaction and the number of interactions that occur in each cell.

Q3.3: very important! at least two hits should have been examined by either over-expression or deletion using CRISPR to reveal the mechanism of tumour growth inhibition and tumour cell killing.

A3.3: The thesis of this paper is that p53 biology in SCLC is impacted by the action of cyclophilins. We show this definitively across 7 main and 18 supplemental figures. While we share the reviewers enthusiasm in understanding the biology of p53-mediated Type D cell death, we believe that systematically targeting each Module 5 gene is beyond the scope of this already large study. Moreover, if we were to select just 2 Module 5 genes to assess their importance for Type D cell death, we would not have a definitive understanding of what Type D cell death is regardless of whether the results were positive or negative. Further, we would not know how to rank the 102 Module 5 genes to select which two to choose. We ask for the reviewer's patience until we can fully decipher the molecular underpinnings of Type D cell death in future work.

REVIEWERS' COMMENTS

Reviewer #1 (Remarks to the Author):

The authors have addressed my 2 major concerns by clearly identifying each primary cell line in representative experiments and by revising their description of the interaction between cyclophilin dependent cell death and the immune system. A possible role for myeloid cells in CsA sensitive cell death has not been fully resolved, but given the reproducibility of the in vitro experiments it would be sufficient to edit the results as follows (or equivalent depending on author style) to emphasize the cell intrinsic nature of this phenomenon.

"Taken together, with in vitro cell culture observations indicating that Type D cells self-destruct in the absence of any immune cells, these data suggest that the immune system is not required to trigger p53-mediated Type D cell death in vivo."

I also would caution the authors that SA-B-gal is a feature associated with cellular senescence but not an unambiguous marker of a very complex cellular state (PMID:26105537). In vitro tests such as colony forming assays are a way to test senescence at the functional level: do arrested cells fail to re enter the cell cycle despite sufficient growth factors. Thus, I would suggest the authors add a qualifier when describing distinctions between reduced proliferation in V versus D cells such as:

"These findings suggest the possibility of inter-tumor heterogeneity wherein p53 reactivation leads to the induction of reduced proliferation with features of cellular senescence in some tumors and necrotic form of cell death in others."

RESPONSE TO REVIEWERS' COMMENTS

ALL REFEREES: We thank the reviewers for their continued interest in our study and their suggestions on how to further improve the quality of our manuscript. We have addressed the remaining concerns below point by point.

REFEREE 1:

Q1.1: The authors have addressed my 2 major concerns by clearly identifying each primary cell line in representative experiments and by revising their description of the interaction between cyclophilin dependent cell death and the immune system. A possible role for myeloid cells in CsA sensitive cell death has not been fully resolved, but given the reproducibility of the in vitro experiments it would be sufficient to edit the results as follows (or equivalent depending on author style) to emphasize the cell intrinsic nature of this phenomenon.

"Taken together, with in vitro cell culture observations indicating that Type D cells self-destruct in the absence of any immune cells, these data suggest that the immune system is not required to trigger p53-mediated Type D cell death in vivo."

A1.1: We have modified the manuscript, as suggested by the reviewer.

Q1.2: I also would caution the authors that SA-B-gal is a feature associated with cellular senescence but not an unambiguous marker of a very complex cellular state (PMID:26105537). In vitro tests such as colony forming assays are a way to test senescence at the functional level: do arrested cells fail to re enter the cell cycle despite sufficient growth factors. Thus, I would suggest the authors add a qualifier when describing distinctions between reduced proliferation in V versus D cells such as:

"These findings suggest the possibility of inter-tumor heterogeneity wherein p53 reactivation leads to the induction of reduced proliferation with features of cellular senescence in some tumors and necrotic form of cell death in others".

A1.2: We have modified the manuscript, as suggested by the reviewer.